# The global spectrum of tree crown architecture

Trees can differ enormously in their crown architectural traits, such as the scaling relationships between tree height, crown width and stem diameter. Yet despite the importance of crown architecture in shaping the structure and function of terrestrial ecosystems, we lack a complete picture of what drives this incredible diversity in crown shapes. Using data from 374,888 globally distributed trees, we explore how climate, disturbance, competition, functional traits, and evolutionary history constrain the height and crown width scaling relationships of 1914 tree species. We find that variation in height–diameter scaling relationships is primarily controlled by water availability and light competition. Conversely, crown width is predominantly shaped by exposure to wind and fire, while also covarying with functional traits related to mechanical stability and photosynthesis. Additionally, we identify several plant lineages with highly distinctive stem and crown forms, such as the exceedingly slender dipterocarps of Southeast Asia, or the extremely wide crowns of legume trees in African savannas. Our study charts the global spectrum of tree crown architecture and pinpoints the processes that shape the 3D structure of woody ecosystems.

Trees come in all shapes and sizes – from incredibly tall and slender, to short with wide, flat crowns[1–7]. This incredible diversity in tree crown architecture plays an important role in driving variation in growth, water use and competition among tree species[1,4,8–10]. Moreover, tree crown architecture underpins key emergent properties of woody ecosystems, including their 3D canopy structure, aboveground biomass, primary productivity and hydrology[8,11–16]. Consequently, uncovering the environmental, ecological and evolutionary drivers that shape the crown architecture of the world's trees is central to better understanding the processes that constrain the structure and function of woody ecosystems. It is also essential for developing more realistic representations of these ecosystems in vegetation models[17–20] and bridging the gap between field and remote sensing observations[21–24] – both of which are needed to track how terrestrial ecosystems are responding to rapid global change.

Differences in crown architecture among trees are the result of species employing a variety of strategies to meet a series of competing physiological, structural, competitive, defensive and reproductive demands (Table 1). Trees expand their crowns vertically and laterally to intercept light, compete with neighbours and disperse seeds, while also needing to maintain water transport to their leaves and mechanical stability[4,6,7,10,25–32]. The balance different tree species strike between these various priorities depends on their environment, ecological strategy and evolutionary history, and will be reflected in the scaling relationships between different axes of tree size, such as their height, crown width and stem diameter[8,9,22,33–41]. For instance, in arid climates woody biomass allocation tends to shift away from height growth and towards crown expansion to limit the risk of hydraulic failure and maximise energy capture, resulting in trees that are shorter for a given diameter and have wider crown profiles[2,8,33,34,38,42–44]. Conversely, when water and nutrients are non-limiting to photosynthesis, strong competition for light leads to greater investment in height growth and relative allocation of carbon to woody tissues[44], pushing trees closer to their structural and hydraulic safety margins[7,8,16,26,27,32,33]. Similarly, tree species have also adapted the size and shape of their crowns to minimise the risk of damage from wind, fire, snow and browsing[40,45–50]. Yet despite clear evidence that crown allometric scaling relationships can vary considerably among tree species, we lack a unified picture of how

✉e-mail: t.jucker@bristol.ac.uk

**Table 1 | Hypothesised drivers of variation in crown architecture among tree species**

| Driver | Key references | Tree height | Crown diameter | Crown aspect ratio |
|---|---|---|---|---|
| Aridity | 2,33,34,42–44,51 | ↓ | ↓ | ↑ |
| Precipitation seasonality | 34,35,38,51 | ↓ | ↓ | ↑ |
| Mean temperature | 34,35,38,42 | ↑ | ↔ | ↓ |
| Aridity × mean temperature | 65,67 | ↓ | ↓ | ↑ |
| Tree cover | 7,16,32,33,59 | ↑ | ↓ | ↓ |
| Maximum wind speed | 27,45,46,70 | ↓ | ↓ | ↓ |
| Fire frequency | 34,40,49 | ↑ | ↑ | ↑ |
| Wood density | 9,36,37 | ↓ | ↑ | ↑ |
| Leaf nitrogen | 38,44,75,76 | ↑ | ↑ | ↔ |
| Specific leaf area | 38,44,75,76 | ↑ | ↑ | ↔ |
| Seed mass | 78–80 | ↔ | ↔ | ↔ |

Predicted relationships between size-standardized estimates of tree height, crown diameter and crown aspect ratio (i.e., after controlling for differences in stem diameter) and various climatic drivers, tree cover (as a proxy for competitive environment), disturbance agents, and functional traits. Upward-pointing arrows in blue (↑) denote positive relationships, while negative ones are shown as downward-pointing red arrows (↓), with the size of the arrows reflecting the expected strength of the relationship. Double-headed arrows (↔) indicate relationships that are expected to be either weak or variable. References supporting each of these hypothesised effects are provided in the table.

and why they do so. Nor do we understand how different axes of crown size and shape covary with one another, how they relate to other key plant functional traits, or how they vary among plant lineages.

Here, we assemble a global dataset capturing information on the stem diameter ($D$), height ($H$), crown diameter ($CD$) and crown aspect ratio ($CAR$, defined as $CD/H$) for over half a million trees (Fig. 1). Using these data, we develop an approach for modelling variation in $H$–$D$, $CD$–$D$ and $CAR$–$D$ scaling relationships among species that allows us to compare their crown sizes and shapes while explicitly controlling for differences in their stem sizes. We apply this method to 1914 well-sampled tree species that span all major woody biomes and clades and use it to: (1) characterise the full spectrum of crown architectural types observed across the world's tree species and biomes; (2) explore whether crown architectural traits are phylogenetically constrained and identify which clades have crown sizes and shapes that are particularly extreme; and (3) test a series of predictions about how $H$–$D$, $CD$–$D$ and $CAR$–$D$ scaling relationships vary in relation to climate, competition, disturbance and other functional traits related to plant metabolism, hydraulics, structural stability and dispersal (Table 1). We show that tree species span a broad range of crown architectural types, encompassing everything from slender to stout stems and narrow to broad crowns. Variation in $H$–$D$ scaling relationships is primarily controlled by water availability and light competition, with tropical forests in Southeast Asia home to disproportionately high concentrations of species with tall, slender growth forms. Conversely, crown width is predominantly shaped by exposure to wind and fire, with legume trees in African savannas achieving some of the widest crowns for their stem diameters.

## Results

### Global variation in crown architectural types
We used data from 374,888 individual trees to generate size-standardised estimates of $H$, $CD$ and $CAR$ for 1914 well-sampled species using two complimentary statistical approaches (see *Methods* for details). The first involved using linear mixed effects models to estimate the height, crown diameter and crown aspect ratio of each species for a tree with a stem diameter of 30 cm (hereafter $H_{D=30}$, $CD_{D=30}$ and $CAR_{D=30}$). The second approach is conceptually and quantitatively similar to the first, but avoids the need to choose an arbitrary stem size at which to compare species (Supplementary Fig. 1). It uses the residuals of a linear regression model to determine if a species has $H$, $CD$ and $CAR$ values that are – on average – larger (positive residuals) or smaller (negative residuals) than expected for their range of stem sizes (hereafter $H_{RESID}$, $CD_{RESID}$ and $CAR_{RESID}$; see Fig. 1e–g for a graphical representation of the method).

Across the 1914 tree species considered in our analysis, we found enormous variation in size-standardized estimates of tree height, crown diameter and crown aspect ratio (Fig. 2). Specifically, $H_{D=30}$ varied 12.1-fold across species, ranging from <4 m in species like *Juniperus osteosperma* and *Maerua crassifolia* to >30 m in several species of the genera *Shorea*, *Parashorea*, *Hopea* and *Vatica* (all Dipterocarpaceae) and as much as 43.2 m in *Eucalyptus regnans*. By contrast, $CD_{D=30}$ was less than half as variable among species, ranging 5.4-fold from <4 m in several species of the genus *Picea* to >14 m in ones like *Brachystegia wangermeeana* and *Pterocarpus tinctorius* in the Fabaceae. As for crown profile shape, $CAR_{D=30}$ ranged 10.5-fold across species. At one end of the spectrum, species like *Abies sibirica* and *Vatica dulitensis* were more than five times as tall as their crowns are wide ($CAR_{D=30} < 0.2$), whilst several species of the genera *Acacia*, *Vachellia* and *Senegalia* (all Fabaceae) had crowns that are noticeably wider than they are tall ($CAR_{D=30} > 1.2$).

For the 1309 species for which we were able to estimate both $H_{RESID}$ and $CD_{RESID}$, we found that these two axes of crown architecture were positively correlated with one another (Pearson correlation coefficient, $\rho = 0.26$, $P < 0.001$). However, despite a general trend of taller trees also having wider crowns, the relationship between $H_{RESID}$ and $CD_{RESID}$ was relatively weak. Species occupied all possible combinations of the tree height *vs* crown diameter spectrum (Fig. 2a), including short and narrow (9.3% of species), short and wide (5.4%), tall and wide (7.9%), and tall and narrow (1.0%).

Where species fell within this spectrum depended, at least in part, on their biome association (Fig. 2b). For example, a high proportion of species found in drylands, temperate woodlands, tropical savannas and tropical dry forests had either short and/or wide crowns (59.6–100% of species, depending on the biome), but almost none were tall (0–8.8%). By contrast, boreal, temperate and tropical rainforests had a considerably higher proportion of species with tall and/or narrow crowns (28.0–60.5%). Overall, differences among biomes were more pronounced for height and crown aspect ratio than for crown width, with biome association explaining 33%, 39% and 5% of the variation in $H_{RESID}$, $CAR_{RESID}$, and $CD_{RESID}$ among species, respectively (see Supplementary Table 4 for pairwise comparisons among biomes based on ANOVAs). However, our results also revealed considerable variation in tree architectural types within biomes (Fig. 2b), highlighting how tree species with very different crown architectures can be found in similar environments.

### Fingerprint of evolution history on crown architecture
From a macroevolutionary perspective, angiosperms and gymnosperms had very similar mean values of $H_{D=30}$ (17.8 m and 17.3 m,

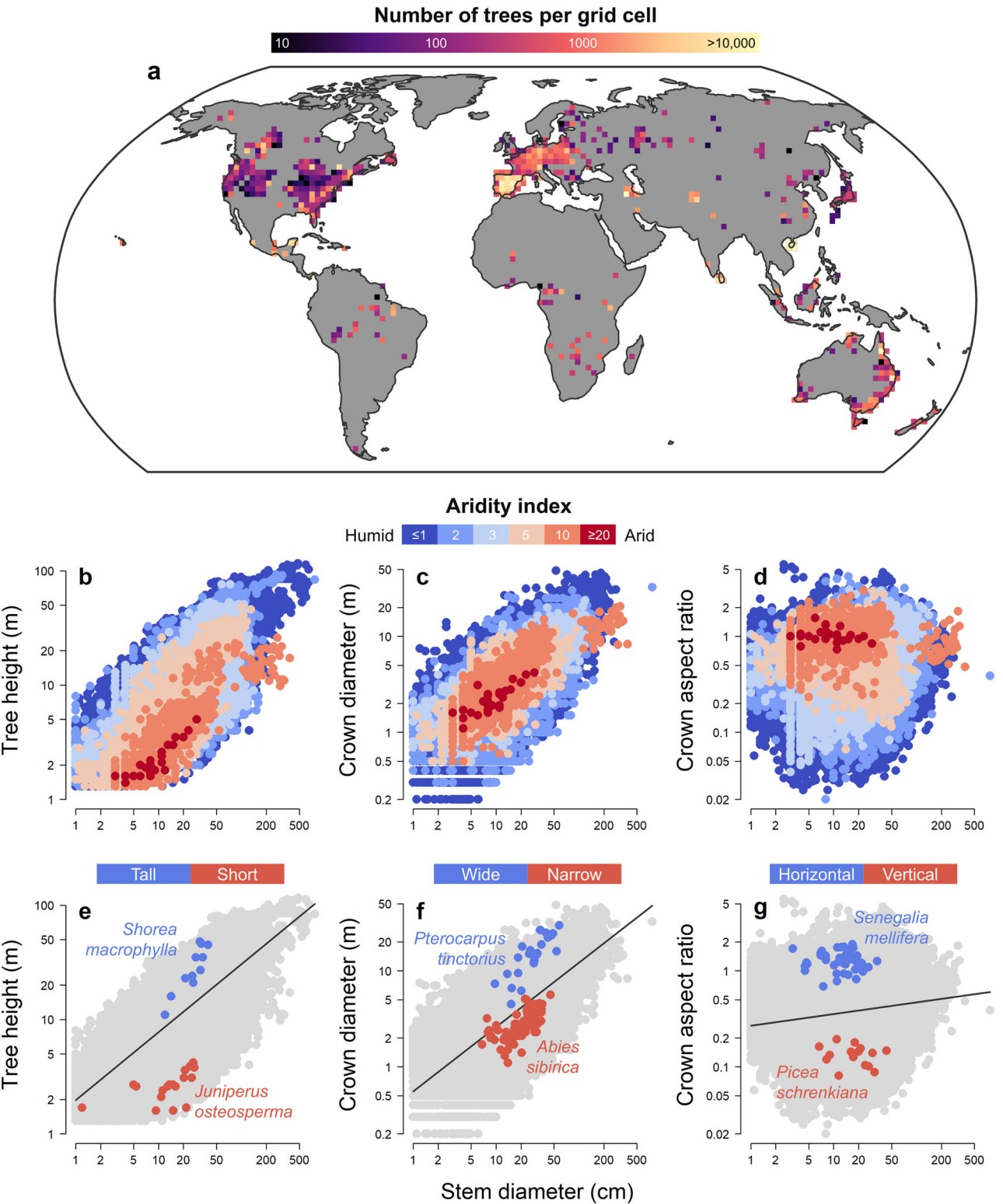

respectively). However, angiosperms spanned a considerably larger range of heights (3.9–42.3 m), particularly at the tall end of the spectrum, where they included 99 of the 100 species with the highest $H_{D=30}$ values. By contrast, both crown diameter and aspect ratio were noticeably larger in angiosperms compared to gymnosperms ($CD_{D=30}$ = 6.4 m $vs$ 5.3 m; $CAR_{D=30}$ = 0.43 $vs$ 0.36). When we placed estimates of $H_{RESID}$, $CD_{RESID}$, and $CAR_{RESID}$ onto a time-calibrated phylogeny of seed plants, we found that all three

exhibited a significant degree of phylogenetic signal (Fig. 3), with Pagel's $\lambda$ values of 0.70, 0.54, and 0.63, respectively ($P < 0.001$ in all cases).

In particular, we found several plant genera and families that stand out based on their tree height, crown size, and shape (Fig. 3 and Supplementary Tables 5–6). For tree height, 25 out of 63 families and 31 out of 86 genera that we tested had mean $H_{RESID}$ values that were significantly different from zero. For angiosperms, species in the

**Fig. 1 | Overview of the global tree allometry database. a** Geographic distribution of the allometric data ($n$ = 374,888 individual trees belonging to 1914 species). Individual tree records were aggregated in 200 × 200 km grid cells (mean number of trees per grid cell = 742). The map was obtained from the Natural Earth database (https://www.naturalearthdata.com) and is displayed using a Robinson projection (EPSG:54030). Relationships between each tree's stem diameter and its **b** height ($H$), **c** crown diameter ($CD$) and **d** crown aspect ratio ($CAR$) are shown on a logarithmic scale. $CAR$ is defined as the ratio between $CD$ and $H$, with values lower than 1 indicating a vertical crown profile ($H > CD$) while values greater than 1 corresponding to a horizontal crown profile ($CD > H$). Points are coloured according to the aridity index value assigned to each tree based its geographic coordinates, with

larger values corresponding to drier conditions (shown in red). Graphical illustration of the approach used to generate size-standardized estimates of **e** tree height ($H_{RESID}$), **f** crown diameter ($CD_{RESID}$) and **g** crown aspect ratio ($CAR_{RESID}$) for each tree species. Regression lines are predicted values obtained by fitting a linear model to the entire dataset (grey points). By comparing predicted and observed value of $H$, $CD$ and $CAR$, we quantified how much each species departs, on average, from this general trend and identified ones with greater (blue points) or smaller $H$, $CD$ and $CAR$ values (red points) than expected given their stem diameters. This approach is conceptually similar to generating species-level predictions of $H$, $CD$ and $CAR$ at a fixed size (e.g., $D$ = 30 cm), but avoids the need to arbitrarily select a size at which to compare species.

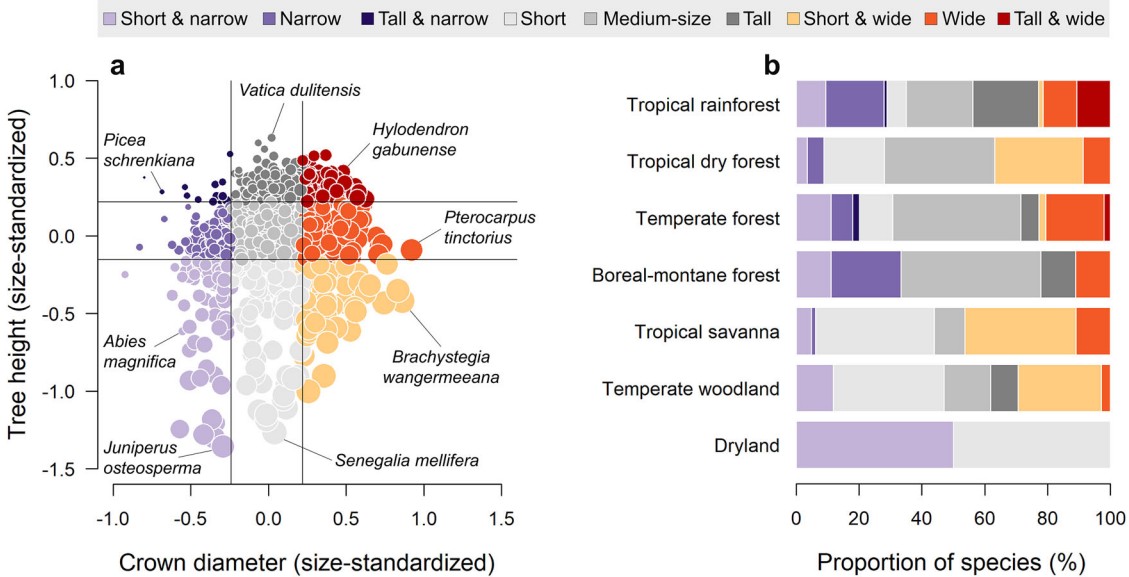

**Fig. 2 | Global spectrum of tree crown architecture.** Tree crown architectural types and their distribution across biomes for the 1309 tree species for which both height and crown size were measured. **a** Tree species were grouped into one of nine architectural types based on their size-standardized height ($H_{RESID}$) and crown diameter values ($CD_{RESID}$). The vertical and horizontal lines mark the 25th and 75th percentile of the data and the size of each circle reflects the crown aspect ratio

($CAR_{RESID}$). Examples of tree species that occupy different areas of this crown architectural spectrum are highlighted. **b** Proportion of species belonging to the nine architectural types for each biome. See Supplementary Table 4 for pairwise comparisons of $H_{RESID}$, $CD_{RESID}$ and $CAR_{RESID}$ values among biomes, and Supplementary Fig. 5 for a breakdown of the nine architectural types among angiosperms and gymnosperms.

Dipterocarpaceae ($H_{D=30}$ = 24.1 m), Myristicaceae ($H_{D=30}$ = 24.4 m), Burseraceae ($H_{D=30}$ = 21.3 m), Annonaceae ($H_{D=30}$ = 21.3 m) and Betulaceae ($H_{D=30}$ = 19.6 m) were particularly tall for their stem diameters (Fig. 3a). From a biogeographic standpoint, we found that Southeast Asia was home to an especially high concentration of species with tall and slender growth forms, with nine of the 10 species with the highest $H_{RESID}$ values native to this region, including species in the genera *Shorea*, *Parashorea* and *Vatica* (Dipterocarpaceae) and *Knema* (Myristicaceae). At the opposite end of the spectrum, species in the Ericaceae ($H_{D=30}$ = 11.4 m), Combretaceae ($H_{D=30}$ = 13.3 m), Fagaceae ($H_{D=30}$ = 15.1 m) and Fabaceae ($H_{D=30}$ = 16.2 m) were significantly shorter that average for a given stem diameter.

The picture within gymnosperms was equally varied. Cupressaceae were generally shorter than expected ($H_{D=30}$ = 14.8 m), despite including species like *Sequoia sempervirens* which can grow incredibly tall in absolute terms. Conversely, within the Pinaceae we found a clear divide between species in the genus *Pinus* which are shorter than average ($H_{D=30}$ = 16.2 m) and those belonging to *Larix* and *Picea* that are taller ($H_{D=30}$ = 22.2 m and 20.6 m, respectively).

In terms of crown size and shape, we found that 21/56 families and 26/60 genera ($CD_{RESID}$) and 12/56 families and 16/60 genera ($CAR_{RESID}$) had values that departed significantly from zero (Supplementary Tables 5–6). One clade that stood out in particular is the Fabaceae (Fig. 3b, c). Of the top 10 species with the highest $CD_{RESID}$ and $CAR_{RESID}$

values, four and six were Fabaceae, respectively. Fabaceae had crowns that are both much wider than average ($CD_{D=30}$ = 7.6 m) and more horizontal in their aspect ratio ($CAR_{D=30}$ = 0.58). This trend was predominantly driven by species that occupy savannas in Africa and the Americas, including ones in the genera *Senegalia*, *Acacia*, *Brachystegia*, *Vachellia* and *Pterocarpus* (mean $CD_{D=30}$ and $CAR_{D=30}$ across 40 tropical savanna specialists = 8.6 m and 0.88, respectively). By contrast, Fabaceae from tropical rainforests ($CD_{D=30}$ = 7.2 m; $CD_{D=30}$ = 0.39) and temperate forests ($CD_{D=30}$ = 6.5 m; $CD_{D=30}$ = 0.37) had crown sizes and profiles that were very similar to other species in these biomes.

Within gymnosperms, species in the Podocarpaceae ($CD_{D=30}$ = 5.2 m), Cupressaceae ($CD_{D=30}$ = 5.3 m) and Pinaceae ($CD_{D=30}$ = 5.3 m) all exhibited narrower than average crowns (Fig. 3b). In the Pinaceae this effect was far stronger than any variation observed in tree height, resulting in crown aspect ratios that are also much smaller than average ($CAR_{D=30}$ = 0.34; Fig. 3c). A similar trend emerged for the Rubiaceae and Lauraceae, where species were generally both tall and with narrow crowns, resulting in low crown aspect ratios ($CAR_{D=30}$ = 0.35 and 0.37, respectively). The exact opposite was true for species in the Combretaceae, Ulmaceae and Fagaceae, which, due to their relatively short stature and wide crowns had particularly large crown aspect ratios ($CAR_{D=30}$ = 0.73, 0.55, and 0.48, respectively).

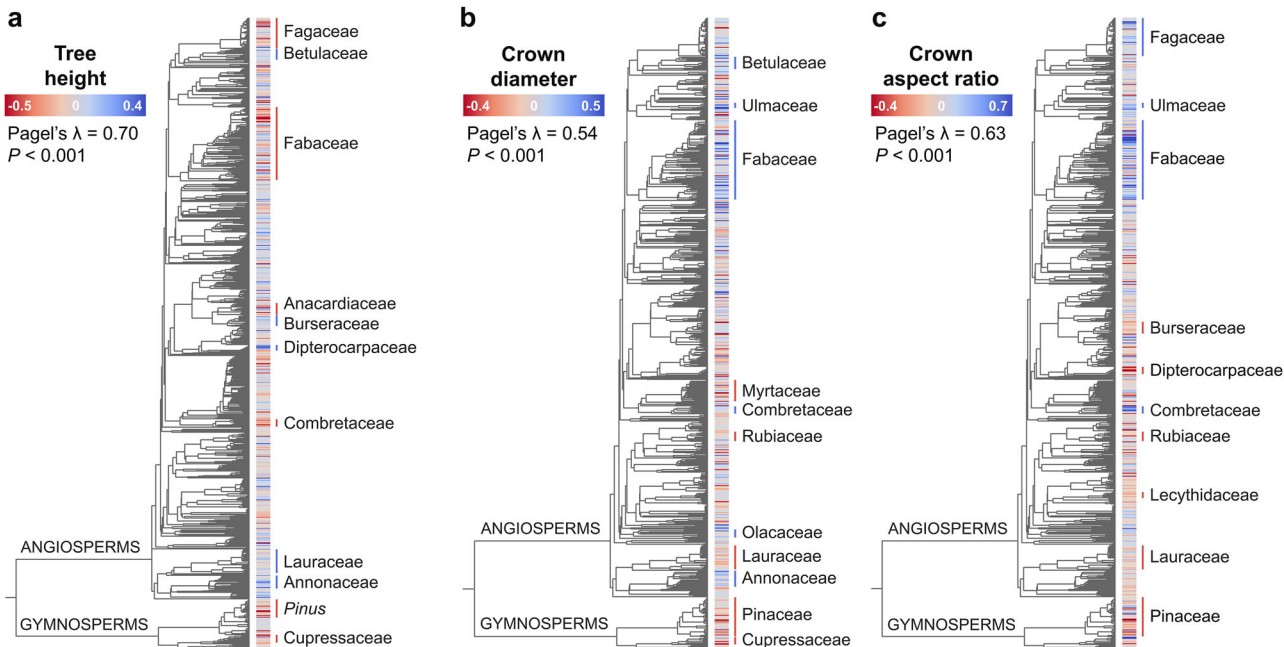

**Fig. 3 | Variation in tree crown architecture across the tree of life.** Size-standardized estimates of **a** tree height ($H_{RESID}$, $n = 1225$ species), **b** crown diameter ($CD_{RESID}$, $n = 870$ species) and **c** crown aspect ratio ($CAR_{RESID}$, $n = 868$ species) are mapped onto a time-calibrated phylogeny of seed plants[116]. Low values (red) indicate species that are shorter, with narrower crowns and smaller crown aspect ratios than expected given the size of their stem, while high values (blue) indicate the opposite. $H_{RESID}$, $CD_{RESID}$ and $CAR_{RESID}$ all exhibited phylogenetic signal, with Pagel's λ values of 0.70, 0.54, and 0.63, respectively ($P < 0.001$ in all three cases based on a likelihood ratio test). Plant families and genera with mean $H_{RESID}$, $CD_{RESID}$ and $CAR_{RESID}$ values that are significantly lower (red lines) or higher (blue lines) than zero are highlighted on each phylogenetic tree (see Supplementary Tables 5–6 for full details). Note that only species that were a direct match to those in the phylogeny were used for the phylogenetic analysis.

## Drivers crown architectural variation in the world's trees

Phylogenetic generalised least squares models relating variation in tree crown architecture among species to climate, tree cover, disturbance, and functional traits explained 55%, 28% and 34% of the variation in $H_{RESID}$, $CD_{RESID}$ and $CAR_{RESID}$, respectively (Fig. 4). Differences in height among species were predominantly controlled by water availability (aridity, and to a lesser extent rainfall seasonality) and tree cover (Fig. 4a), with $H_{RESID}$ decreasing rapidly with rising aridity and increasing steadily as tree cover increased (Fig. 5a, b). On average, species growing where potential evapotranspiration was equal to or less than mean annual precipitation (aridity index ≤ 1) were almost twice as tall for a given stem diameter ($H_{D=30} = 20.1$ m) as those where the aridity index was ≥2 ($H_{D=30} = 11.1$ m). Similarly, $H_{D=30}$ increased from 11.5 m to 21.4 m when comparing species growing where tree cover was ≤20% and ≥80% (Fig. 5a). However, we also found that when water was non-limiting (aridity index ≤ 1), trees could vary hugely in their investment in height growth, with $H_{D=30}$ ranging anywhere between 6.4 m and 43.2 m (Fig. 5b). In contrast to aridity and tree cover, we only observed a modest positive association between $H_{RESID}$ and mean annual temperature (MAT). Temperature did however indirectly influence $H_{RESID}$ through its interaction with aridity (Supplementary Fig. 6). Specifically, we found that $H_{RESID}$ declines with increasing aridity was much less pronounced for trees growing in cold climates (MAT < 10 °C) compared to warm ones (MAT > 20 °C).

Water availability and tree cover also emerged as significant predictors of $CD_{RESID}$, with species generally having narrower crowns in more arid and seasonal climates, and where tree cover was higher (Fig. 4b). But these effects were much less pronounced than for tree height (Fig. 5d, e). Consequently, we found that variation in $CAR_{RESID}$ along aridity and tree cover gradients was mostly driven by changes in $H_{RESID}$, with crown profiles becoming markedly more vertical as aridity decreased and tree cover increased (Figs. 4c, 5g, h). For example, the average $CAR_{D=30}$ of species growing where tree cover was ≤20% (0.77)

was more than twice that of those where tree cover was ≥ 80% (0.36). Conversely, crown diameter was much more strongly influenced by disturbances such as wind and fire (Fig. 4b). In particular, $CD_{RESID}$ decreased markedly as maximum wind gust speeds increased (Fig. 5f). Species also tended to be shorter for a given stem diameter where wind speeds were higher (Fig. 5c), but this effect was more subtle than for crown diameter, meaning that overall crown profiles became significantly narrower as wind speeds increased (Fig. 5i). By contrast, areas with higher frequencies of wildfires harboured species with wider crowns and higher crown aspect ratios (Fig. 4b, c), but with similar $H_{RESID}$ values.

While variation in $H_{RESID}$, $CD_{RESID}$, and $CAR_{RESID}$ among tree species was predominantly associated with climate, tree cover and risk of disturbance, we also found that these crown architectural traits covaried with other plant functional traits (Fig. 4). After accounting for environmental effects, we found that both $H_{RESID}$ and $CD_{RESID}$ were significantly greater in species with higher leaf nitrogen content, with $H_{RESID}$ also positively correlated to specific leaf area. Additionally, while we observed no clear relationships between $H_{RESID}$ and wood density, we did find that species with wider crowns and more horizontal crown profiles had denser wood. By contrast, none of the three crown architectural traits exhibited any relationship with seed mass.

## Discussion

Using allometric data from hundreds of thousands of trees across the world, our study provides a global picture of how tree species vary in their crown architecture (Fig. 2) and what drives this variation (Fig. 4). While ecologists have long been aware that trees can differ in their crown size and shape[1,2,5,7,10], we have lacked a quantitative understanding of where the boundaries of this crown architectural spectrum lie. Not only did we show that tree species can vary considerably in the scaling relationships between their height, crown width, and stem diameters, but we also found that size-standardised estimates of tree

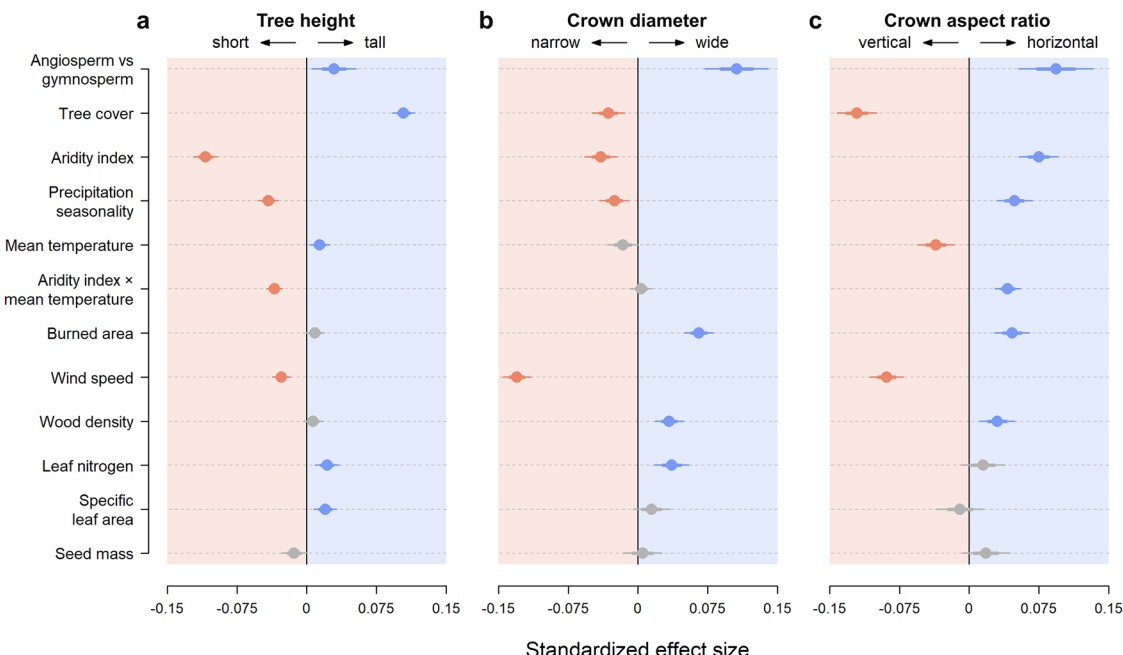

**Fig. 4 | Drivers of variation in crown architecture among tree species.** Standardized model coefficients for each predictor variable were obtained by fitting phylogenetic generalised least squares regressions to size-standardized estimates of **a** tree height ($H_{RESID}$, $n = 1910$ species), **b** crown diameter ($CD_{RESID}$, $n = 1313$ species), and **c** crown aspect ratio ($CAR_{RESID}$, $n = 1309$ species). Error bars show both standard errors (thick lines) and 95% confidence intervals (thin lines) of the model coefficients. Significantly positive and negative coefficients are shown in blue and red, respectively, while those for which the 95% confidence intervals overlap with zero are shown in grey.

height and crown diameter were largely decoupled, forming two independent axes of variation in crown architecture (Fig. 2a).

Where species fell within this crown architectural spectrum depended largely on their environment, their evolutionary history, and ecological strategy. Like previous studies, we found that species confined to more arid and seasonal biomes were generally much shorter for a given stem diameter than those growing in more humid climates[2,8,38,44,51]. We also found that the range of crown architectural types was much greater in biomes where water and temperature were non-limiting and angiosperms dominate the flora, such as tropical rainforests and temperate forests. In these environments, tree species living side by side can have incredibly different crown forms depending on their ecological strategy and evolutionary history[8,36,52–55], whereas where conditions for growth are harsher, there is less flexibility in the range of crown sizes and shapes that species can assume.

From a macroevolutionary standpoint, when growing in similar environments angiosperm were only marginally taller for a given diameter than gymnosperms. However, angiosperms consistently had wider crowns. This pattern reflects a fundamental difference in the growth strategy of the two major clades, with gymnosperms investing less in lateral crown expansion due to strong apical dominance and control[56], resulting in crown profiles that are generally more vertical than those of angiosperms[22,33]. When exploring how evolutionary history has shaped variation in crown architecture in more detail, we found that $H_{RESID}$, $CD_{RESID}$, and $CAR_{RESID}$ all exhibited a clear phylogenetic fingerprint (Fig. 3), with several plant linages standing out. For instance, we found that dipterocarps – and trees native to Southeast Asia more generally – achieve remarkable heights for a given stem diameter[39,40,57]. In terms of crown size, we showed that several species of Fabaceae that grow in tropical savannas in Africa and the Americas had exceptionally wide crowns. This helps explain previous observations that trees in these regions have larger crowns than those of Australia[8,34,40], where savannas are largely dominated by smaller-crowned eucalypts.

Aridity and tree cover emerged as the strongest predictors of tree height scaling relationships, which is consistent with previous empirical and theoretical research showing how investment in height growth is modulated by risk of hydraulic failure[2,17,30,42–44,58] and competition for light[7,16,32,33,59]. Interestingly, species that were tallest for a given stem diameter ($H_{D=30} > 30$ m, top 1% of species) occurred within a narrow band of aridity values (0.65–0.81) where rainfall only slightly exceeded potential evapotranspiration (Fig. 5b). Once the aridity index decreased below 0.5, species with very high $H_{RESID}$ values disappeared, possibly due to a combination of waterlogging from excessive rainfall and/or growth limitations linked to lower temperatures and high cloud cover[60–62].

$CD_{RESID}$ also decreased with aridity, suggesting trees limit the size of their crowns (and by proxy their total leaf area) in environments where water is scarce to reduce transpiration and minimise risk of hydraulic stress[8,33,34,63]. As expected, we also found that $CD_{RESID}$ decreased with tree cover, which is consistent with trees prioritizing height growth over crown expansion in response to increasing competition for light[7,16,32–34]. However, the effects of both aridity and tree cover were much weaker for crown diameter than for height. One possible reason for this is that tree cover and aridity are generally negatively correlated, leading to their effects cancelling each other out: in humid climates where tree cover also tends to be high, trees would be able to support wider crowns, were it not for increased competition for space. The net result is that investment in crown expansion relative to height growth (as captured by $CAR_{RESID}$) increased progressively in drier and more open habitats (Fig. 5g, h), a growth strategy that maximises energy capture and hydraulic safety[44].

In contrast to aridity and tree cover, temperature played a secondary role shaping variation in tree architecture (Fig. 4). However, we did find that trees tended to be taller for a given diameter in warmer climates, which is consistent with observations that the world's tallest trees inhabit mild and warm climates with little seasonality[62]. This also fits our understanding of how cold temperatures impact tree height growth[64] and the fact that trees in cold climates generally have small

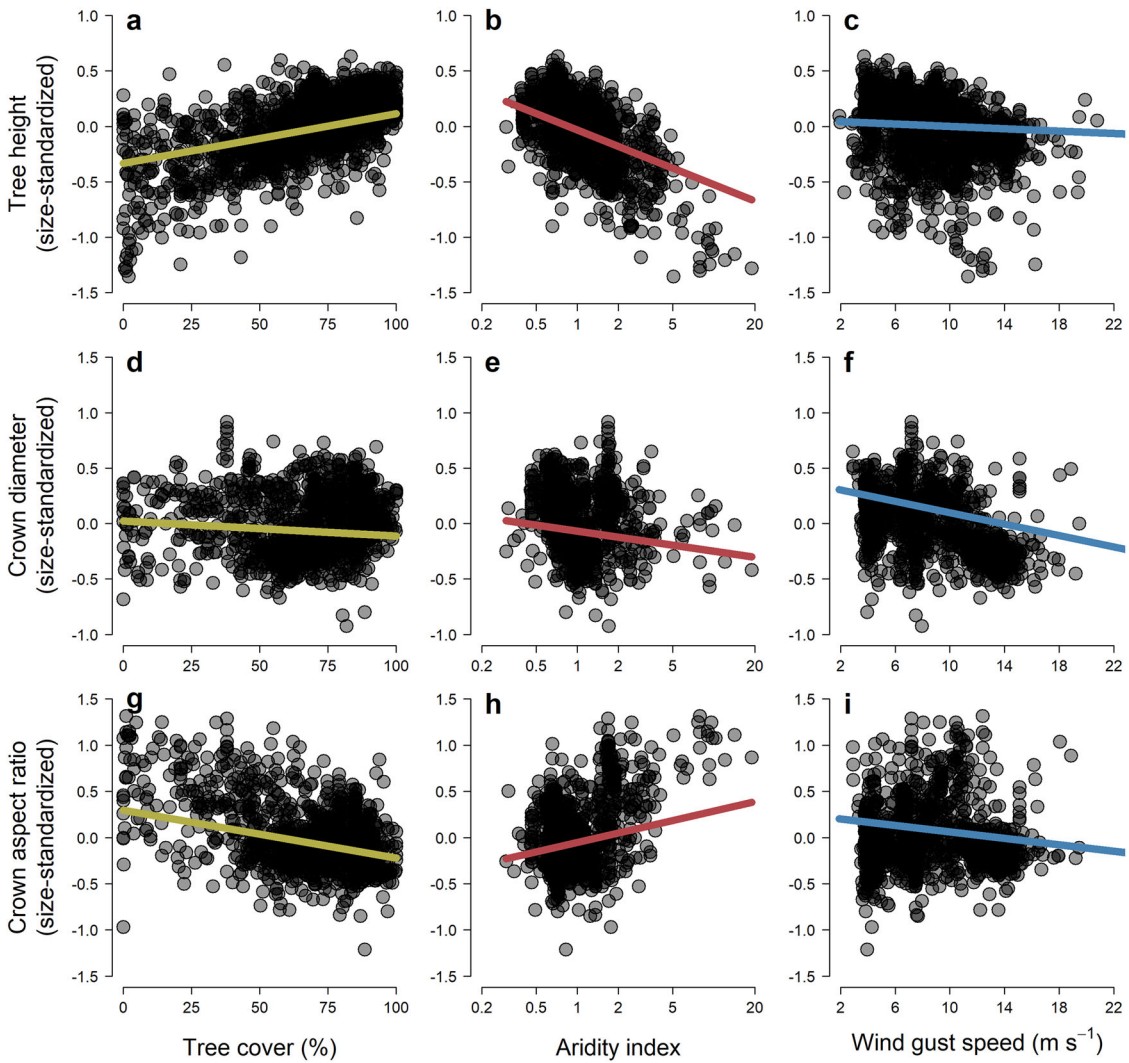

**Fig. 5 | Variation in the crown architecture along gradients of tree cover, aridity, and wind gust speed.** Points are species-level estimates of size-standardized **a**–**c** tree height ($H_{RESID}$, $n = 1910$ species), **d**–**f** crown diameter ($CD_{RESID}$, $n = 1313$ species) and **g**–**i** crown aspect ratio ($CAR_{RESID}$, $n = 1309$ species). Fitted lines correspond to phylogenetic generalised least squares model predictions generated by keeping all other predictors fixed at their mean values. Negative values of $H_{RESID}$, $CD_{RESID}$, and $CAR_{RESID}$ indicate species that are shorter, with narrower crowns and smaller crown aspect ratios than expected given the size of their stem, while positive values denote the opposite. Values of tree cover, aridity index and wind gust speed represent are means calculated across all individual trees of a given species. Note that the aridity index was log-transformed and that larger values correspond to drier conditions.

vessels to minimise risk of embolisms under freezing conditions, which limits their ability to grow tall[42]. Moreover, we found that temperature indirectly influenced tree height scaling relationships by exacerbating the effects of aridity (Supplementary Fig. 6). Warmer temperatures are associated with higher vapour pressure deficits, requiring trees to have more soil water to meet higher evaporative demands[65–67]. This suggests that even small decreases in rainfall and/or increases in temperature in warm climates could lead to disproportionately large impacts on forest structure[2,68,69].

Risk of disturbance by wind and fire emerged as stronger predictors of crown diameter scaling relationships than climate (Fig. 4). Species adapted to windy conditions were shorter, had narrower crowns, and more vertical crown profiles for a given stem diameter than those where risk of exposure to high wind speeds was low. These adaptations would make them less prone to uprooting and snapping in high winds, as the risk of both is proportional to total crown surface area[27,31,45,46]. Our findings are also consistent with observations that some of the world's tallest tropical trees grow where the risk of wind disturbance is low, such as the Guiana Shield and in Borneo[27,31,39,70].

Crown width was also positively associated with burned area fraction, indicating that trees in fire-prone environments generally allocate more resources to lateral crown expansion. This could partly be a result of lower competition for light in more open environments where fire is frequent, which would allow trees to maximise light interception by spreading their crowns laterally[40]. It could also reflect a more direct response to fire, with trees developing wide crowns to shade out grasses and limit fuel loads[71]. Additionally, adaptations to fire may be confounded with those associated with herbivory, which also plays an important role in shaping tree architecture in savannas[49,50], with wider crowns serving as a protective strategy against browsing[40,72]. In terms of how fire might affect height scaling relationships, our expectation was that trees would generally invest more in height growth to escape fire (and herbivory). By contrast, we found no relationship between $H_{RESID}$ and burned area fraction, which could be because some species adopt the alternative strategy of fire resistance through the growth of thicker stems and bark[40,49].

As for other disturbance agents, such as snow accumulation, we would expect trees exposed to high snow loads to have narrower crowns and more slender profiles[47,48]. While we did not test this

directly, we did find that $CAR_{RESID}$ was generally lower in colder climates. Given that snow cover duration and mean annual temperature were highly correlated (Supplementary Fig. 3), this temperature response may in part reflect an adaptation to minimising the risk of stem breakage from snow accumulation in the crown in cold climates.

Crown architectural traits covaried with several other plant functional traits related to photosynthesis and structural integrity. For instance, we found that $CD_{RESID}$ and $CAR_{RESID}$ were positively associated with wood density, which is thought to confer the mechanical strength and resistance needed for trees to grow large branches and wide crowns[9,36,37]. However, while our results support this hypothesis, we also found that numerous species had wide, horizontal crowns despite having relatively low wood densities. This highlights how other properties, such as branching architecture, may be just as important in determining a tree's structural integrity[27].

Species that were taller and with wider crowns for a given stem diameter also generally had higher concentrations of nitrogen in their leaves. Leaf nitrogen content is a cornerstone of the 'fast-slow' plant economic spectrum[73,74], with species that have high leaf nitrogen generally capable of rapid growth, but also less able to tolerate shade due to higher metabolic and respiration rates[75,76]. Based on this, we would expect species with higher leaf nitrogen to invest more in both height growth and crown expansion to allow them to intercept more light, limit self-shading, and optimise the distribution of leaves across their crowns. Moreover, higher leaf nitrogen content should lead to greater photosynthetic rates and investment in height growth, as found previously along a rainfall gradient in Australia[44]. This is also consistent with work from the U.S. showing that taller and more slender trees tend to be less shade tolerant[38]. Similarly, early successional species in tropical forests that are adapted to grow rapidly in height to take advantage of gap openings generally have high leaf nitrogen content[52,77].

By contrast, we found no evidence that large-seeded species were architecturally any different to those with small seeds. Seed mass has previously been shown to correlate positively with maximum plant height and canopy area, which some have proposed is the result of species with longer life spans and greater adult sizes being more likely to invest in large seeds[78-80]. Alternatively, seed mass may correlate with crown architecture through its association with seed dispersal[28,81]. For example, species with light, wind-dispersed seeds might profit from being taller to increase dispersal range, while species that have ballistic seeds or ones that simply drop to the ground might benefit from wider branches to increase distance from parent trees[52,81]. However, we found no support for either of these hypotheses when relating variation in seed mass to $H_{RESID}$ and $CD_{RESID}$. If these processes are at play, they may well be better captured by other facets of crown architecture (e.g., maximum tree height and crown width).

Our findings highlight several fruitful avenues for future research. An obvious next step would be to expand the spectrum of crown architectural traits to other axes of crown size and shape, such as crown depth, surface area, and volume[8,16,34]. This would allow us to test long-standing predictions about how crown size and shape reflect a compromise (in terms of carbon gains) between greater light interception and higher maintenance costs[6], and explore how the outcome of these trade-offs varies with water and light availability[8,82]. In this regard, efforts to better characterise crown architecture are likely to benefit from growing access to technologies such as terrestrial laser scanning (TLS). These can provide a much richer picture of a tree's crown and local surroundings, including reconstructing its branching structure, accurate 3D volumes and within-crown distribution of leaves[81,83,84]. Extracting these measurements from TLS point clouds remains a challenge, but access to data and automated processing pipelines are continuously improving[1,85].

In addition to extending the range of crown architectural traits, another important direction for future research would be to explore how crown allometric scaling relationships vary within species. Our study focused exclusively on species-level differences in crown architecture, as for most species we had insufficient data to robustly incorporate intraspecific variation. However, it is well known that trees can exhibit considerable plasticity in their crown shapes and sizes, shifting their allocations to vertical and horizontal growth as they age and in response to both competition and climate[2,4,16,33]. What remains less clear is the extent to which shifts in crown architecture along environmental gradients observed across species are mirrored by ones occurring within species. A previous study of 342 widely-distributed species suggests that shifts in $H–D$ scaling relationships along aridity gradients reflect both turnover in species composition and intraspecific plasticity[2]. However, it also revealed that patterns across and within species were only consistent in 70% of cases, and that intraspecific plasticity was of secondary importance (relative to species turnover) in driving shifts in $H–D$ scaling relationships along aridity gradients[2]. Moreover, this analysis focused exclusively on tree height, and much less is known about plasticity in crown size and shape outside of a few temperate forest species[16,33]. A big question concerns the relative importance of ontogeny, competition, and the environment in driving plasticity in crown architecture. Similarly, it is unclear whether certain plant lineages show greater ability to plastically adapt the size and shape of their crowns compared to others, and if so why.

While there is clearly plenty of scope to build on our findings, our study provides a key starting point for characterising the crown architectural spectrum of the world's trees. Not only do we capture the range of possible crown architectural types, we also take an important step towards explaining what drives this immense variation. Our results highlight how crown architecture is jointly constrained by a range of processes related to a tree's environment, ecological strategy, and evolutionary history. This understanding will underpin ongoing efforts to leverage remote sensing technologies to track tree carbon stocks and dynamics at scale[21-24]. It is also critical for developing the next generation of Earth System Models that accurately simulate variation in vegetation structure and dynamics by incorporating more realistic representations of how tree crowns vary among biomes, plant functional types, and in coordination with other traits[17,19,20]. All of this is essential to better understanding the processes that shape the structure and function of woody biomes and tracking how these are responding to rapid global change.

## Methods
### Individual tree height and crown size data
We compiled 528,311 georeferenced records of individual trees for which stem diameter ($D$, cm), height ($H$, m) and/or crown diameter ($CD$, m) were measured (Fig. 1a). For trees where both $H$ and $CD$ were measured ($n = 340,221$; 64.4%), we also calculated their crown aspect ratio ($CAR$) as $CD/H$, where $CAR < 1$ denotes a vertical crown profile and $CAR > 1$ a flat or horizontal profile[1,86]. These data were obtained from 62,435 globally distributed sites which encompass all major terrestrial biomes and span a gradient in mean annual temperature of $−15.1–30.1\,°C$ and $143–7157\,mm\,yr^{-1}$ in rainfall. Sampled trees span multiple orders of magnitude in size and crown shape (Fig. 1b–d) and represent 5161 tree species from 1451 genera and 187 plant families.

Most of the data (94.4% of records) were sourced from the Tallo database[2]. Additionally, we also obtained data from Alberta's Permanent Sample Plots network in Canada ($n = 12,171$ trees) and the ICP Forests network in Europe ($n = 17,540$ trees). Allometric data were quality controlled following the protocols of the Tallo database[2]. Briefly, we first used Mahalanobis distance to identify and remove possible data entry errors by screening for trees with unrealistically large or small $H$ and $CD$ values for a given stem diameter. Species names were then standardized against those of The Plant List (TPL) using the *taxonstand* package[87] in R (version 4.2.2)[88]. Lastly, we excluded records from species that did not meet our working

definition of trees: perennial woody seed plants with a single dominant stem that are self-supporting and undergo secondary growth (i.e., excluding ferns, palms, short multi-stemmed shrubs, and lianas).

## Species level, size-standardized estimates of tree height, crown diameter, and crown aspect ratio

Tree species can differ considerably in their maximum size and developmental strategies, so to directly compare their crown architecture, we used two complimentary approaches to generate size-standardized estimates of $H$, $CD$, and $CAR$ at the species level. For these and all subsequent analyses, we focused on species with at least 10 trees sampled within the same biome and spanning a minimum $D$ range of 20 cm between the smallest and largest measured tree (see *Environmental data* for details on how trees were assigned to biomes). In total, 1914 species represented by 374,888 individual trees met these criteria for at least one of the three axes of crown size and shape and 1309 species represented by 251,733 trees had sufficient data for all three (1910 species for $H$, 1313 for $CD$ and 1309 for $CAR$; see Supplementary Table 1 for details). These 1914 species cover 755 genera and 131 plant families.

The first approach to comparing species' crown architectures involved generating estimates of $H$, $CD$, and $CAR$ for a tree of fixed size ($D = 30$ cm) for each species[33,52]. To do this, we modelled variation in $H$, $CD$ and $CAR$ among individual trees as a power-law function of $D$ by fitting linear mixed-effects regressions to log-log transformed data[8,33,34]:

$$\log(Y) = \alpha_{f/g/s} + \beta_{f/g/s} \log(D) \quad (1)$$

where $Y$ denotes either $H$, $CD$ or $CAR$, $\alpha$ is the intercept (or normalization constant) and $\beta$ is the slope (or scaling exponent). Models were fit using the *lme4* package[89] and both the intercept and slope of the regressions were allowed to vary among tree species, genera, and families with a nested random effects structure (denoted by the $f/g/s$ subscripts in the equation above). $R^2$ values accounting for both fixed and random effects components of the models were 0.82, 0.74, and 0.45 for $H$, $C$,$D$ and $CAR$, respectively[90]. The fitted models were then used to predict $H$, $CD$, and $CAR$ for each specie,s assuming a fixed stem size of 30 cm (hereafter $H_{D=30}$, $CD_{D=30}$ and $CAR_{D=30}$). Models were fit using least squares regression (as opposed to approaches such as major axis regression) as we were not interested in comparing scaling coefficients among species, but in using the fitted models to generate predictions[91]. For the $H$–$D$ model, we compared the power-law with a saturating Michaelis-Menten function, as previous work has suggested that the latter might fit the data better[37]. We found that both approaches yielded very similar estimates of $H_{D=30}$ ($\rho = 0.93$), but that overall the power-law provided a better fit to the data (Supplementary Fig. 2).

The second approach we developed to capture variation in crown architecture among species is conceptually similar to the first, but avoids the need to choose an arbitrary stem size at which to compare species[8]. As for the previous method, we began by using the tree-level data to model variation in $H$, $CD$ and $CAR$ as a power-law function of $D$. However, in this case we explicitly ignored differences among species and simply estimated the overall scaling relationships between $H$–$D$, $CD$–$D$ and $CAR$–$D$ across the whole dataset by fitting ordinary linear regressions to log-log transformed data. Using the residuals of the models (i.e., the difference between observed and predicted values of $H$, $CD$ and $CAR$ on a log-log scale), we then determined *post hoc* whether a given species has $H$, $CD$ and $CAR$ values that are – on average – larger (positive residuals) or smaller (negative residuals) than expected after accounting for differences in stem size between trees (see Fig. 1e–g for a graphical representation of this approach). Because sample sizes varied considerably among species ($n = 10$–22,835 trees per species), we subset the data by randomly selecting 10 trees per

species prior to model fitting. Without this step, well-sampled species would dominate the signal of the regression and skew the values of the residuals. This randomization step was repeated 100 times, and for each species we then calculated the mean value of the residuals across all model runs as a measure of size-standardized $H$, $CD$, and $CAR$ (hereafter $H_{RESID}$, $CD_{RESID}$, and $CAR_{RESID}$).

Quantitatively, the two approaches gave very similar results (Supplementary Fig. 1). However, the second method based on model residuals is better suited to comparing species that exhibit contrasting growth trajectories or vary in their size at maturity, as it integrates data across all observed tree sizes instead of focusing on a single point of comparison (e.g., $D = 30$ cm, which could correspond to a small tree for some species and a very large one for others). For subsequent analyses, we therefore focus on comparing values of $H_{RESID}$, $CD_{RESID}$, and $CAR_{RESID}$ across species, but to aid the interpretation of results, we also report values of $H_{D=30}$, $CD_{D=30}$ and $CAR_{D=30}$. Note that both of these approaches overlook plasticity in allometric scaling relationships within species. This is partly because previous work based on the Tallo database has shown that allometric variation along environmental gradients is twice as pronounced among species as it is within them[2] – prompting us to focus on identifying drivers of species-level differences in crown architecture. But it also reflects the fact that for many species in our analysis we have insufficient data to appropriately model intraspecific variability (314 species were measured at a single site and <20% were sampled widely enough to characterise their architectural plasticity along environmental gradients[2]).

## Environmental data

To understand how environmental conditions shape variation in $H_{RESID}$, $CD_{RESID}$ and $CAR_{RESID}$ among tree species, we used the geographic coordinates of individual trees to assign attributes related to climate, competition, and disturbance (see Supplementary Table 2 for full details on sources of environmental data). These environmental predictors were chosen based on previous work suggesting they play an important role in shaping tree crown allometry by constraining plant hydraulics, growth and competition (Table 1). Importantly, they were also selected as they were not strongly correlated with one another (Supplementary Fig. 3), allowing their effects to be teased apart in subsequent analyses. In addition to the environmental predictors described below, trees were also assigned to one of seven biome classes based on the classification used by the Terrestrial Ecoregions of the World database[92].

For climate, we focused on the effects of mean annual temperature (MAT, °C), precipitation seasonality (mm,) and aridity (unitless index). MAT and precipitation seasonality were obtained from the WorldClim2 database at a resolution of 30 arc-seconds[93]. Aridity was instead calculated as the ratio between potential evapotranspiration (PET, mm) and mean annual precipitation (MAP, mm), where MAP was obtained from WorldClim2 while PET was derived from the Global Aridity Index and Potential Evapotranspiration Climate Database at 30 arc-second resolution[94]. This is the inverse of how aridity is often expressed[94], but has the advantage of being easier to interpret as larger values of the aridity index correspond to drier conditions[95].

As a proxy for local competitive environment, we used estimates of tree cover derived from MODIS at 15-arc second resolution for the year 2008[96], which broadly overlaps with the period when the majority of the allometric data were collected. We chose this approach as for most trees we lacked information on stand-level attributes commonly used to characterise competition, such as basal area or stem density[16,33]. However, for a subset of sites across which these field data were available, we found good agreement between MODIS-derived estimates of tree cover and stand basal area (Supplementary Fig. 4), suggesting satellite estimates of tree cover provide a reliable indicator of competitive environment.

As indicators of disturbance, we focused on wind speed and fire risk. Specifically, we used the ERA5-Land data to calculate the maximum wind gust speed experienced by each tree between 2010–2020. To quantify exposure to fire, we calculated the mean burned area fraction between 2001–2010, as estimated from MODIS in the Global Fire Emissions Database[97]. Note that we also tried to assess the impacts of snow accumulation on tree crowns, but found that estimates of snow cover duration derived from MODIS were strongly correlated with MAT ($\rho = -0.80$; Supplementary Fig. 3), and were therefore not considered in subsequent analyses.

For each of the 1914 species that met the minimum sampling criteria described previously, we then calculated mean values of aridity, MAT, precipitation seasonality, tree cover, maximum wind gust speed and burned area fraction across all sampled trees. Each species was also assigned to a unique biome based on the terrestrial ecoregion in which they were recorded most frequently[98].

## Functional trait data

To test how variation in $H_{RESID}$, $CD_{RESID}$ and $CAR_{RESID}$ among tree species relates to other key plant functional traits, we compiled data on wood density (g cm$^{-3}$), leaf nitrogen content (mg g$^{-1}$), specific leaf area (SLA, mm$^2$ mg$^{-1}$) and seed mass (g) from multiple sources (Supplementary Table 3). This includes the TRY plant trait database[99], the Botanical Information and Ecology Network (BIEN) database[100], the global wood density database[101], the Royal Botanic Gardens Kew seed information database, the AusTraits database[102], the China plant trait database[103], the Terrestrial Ecosystem Research Network (TERN), as well as selected publications[36,77,104–107]. These four functional traits were chosen as previous work suggests they may covary with crown size and shape through their influence on whole-plant growth, size, hydraulics, and mechanical stability[33,36,37,75,76,79] and have been measured for numerous tree species.

To obtain species-level mean values for each trait, we first grouped together individual records by site (based on shared geographic coordinates) and then species[108]. For species where no individual-level records could be sourced, species-level values reported in the literature were used instead if available. Of the 1914 tree species for which we estimated $H_{RESID}$, $CD_{RESID}$, and/or $CAR_{RESID}$, we obtained wood density estimates for 1572 species (82%), leaf nitrogen content for 1085 (57%), SLA for 1120 (59%), and seed mass for 1108 (58%).

## Mapping the spectrum of crown architectural types and its distribution across woody biomes

To characterise the range of crown forms that tree species can assume and better understand how these vary among woody biomes, we used estimates of $H_{RESID}$, $CD_{RESID}$ and $CAR_{RESID}$ to determine how species cluster into architectural types based on their height, crown size, and shape[1]. To do this, we first calculated the correlation between $H_{RESID}$ and $CD_{RESID}$ to determine how tightly constrained these two axes of crown architecture are across the 1309 species where both had been measured. A strong positive correlation would indicate that species that are taller for a given stem diameter also tend to have larger crowns and vice versa. Conversely, a weak correlation between $H_{RESID}$ and $CD_{RESID}$ would suggest that when standardized by size, tree species are able to adopt a wide range of crown architectural forms, from tall and narrow to short and wide.

We then grouped species into one of nine crown architectural types: (1) short and narrow, (2) narrow, (3) tall and narrow, (4) short, (5) medium-sized, (6) tall, (7) short and wide, (8) wide, and (9) tall and wide species. Species were assigned to groups based on their $H_{RESID}$ and $CD_{RESID}$ values and whether these fell in the lower quartile, interquartile range, or upper quartile of data (see Fig. 2a for a visual representation). Note that $CAR_{RESID}$ was not used to group species, as

any differences in $CAR_{RESID}$ among species can be directly attributed to ones in $H_{RESID}$ and $CD_{RESID}$. To determine the degree to which crown forms are adapted and confined to specific environments, we quantified the relative frequency of each architectural type across different biomes. To support this analysis, we also used one-way ANOVAs to compare mean values of $H_{RESID}$, $CD_{RESID}$, and $CAR_{RESID}$ among species from different biomes.

## Evolutionary history and its fingerprint on crown architecture

To determine whether crown architectural traits exhibit phylogenetic signal, we mapped $H_{RESID}$, $CD_{RESID}$, and $CAR_{RESID}$ onto the Smith & Brown (2018) phylogeny of seed plants and calculated Pagel's $\lambda$ as a general test of phylogenetic signal for each crown attribute[109,110]. A $\lambda$ value of 0 indicates no phylogenetic signal, while a value of 1 corresponds to a trait that has evolved according to Brownian motion, indicating strong phylogenetic signal[109]. Pagel's $\lambda$ was calculated using the *phytools* package[111], which uses a likelihood ratio test to determine whether $\lambda$ is significantly different from 0. Because tests of phylogenetic signal are sensitive to errors in the phylogeny, such as those associated with branch lengths[110], only species that were a direct match to those in the time-calibrated phylogeny were retained for subsequent analyses (1225 species for $H_{RESID}$, 870 for $CD_{RESID}$ and 868 for $CAR_{RESID}$; Supplementary Table 1).

To complement $\lambda$ – which provides a global test of phylogenetic signal across the entire phylogeny – we also explored how $H_{RESID}$, $CD_{RESID}$ and $CAR_{RESID}$ varied among clades within the phylogeny. Specifically, we used one-way ANOVAs fit without an intercept to identify plant families and genera where species' mean $H_{RESID}$, $CD_{RESID}$ and $CAR_{RESID}$ values are significantly greater or smaller than zero. For this purpose, we only retained families and genera represented by at least five species in our dataset ($n = 63$ families and 86 genera for $H_{RESID}$, and $n = 56$ families and 60 genera for $CD_{RESID}$ and $CAR_{RESID}$).

## Effects of climate, competition, disturbance and functional traits on crown architecture

To quantify the effects of climate, competition, and disturbance on tree crown architecture, we modelled variation in $H_{RESID}$, $CD_{RESID}$ and $CAR_{RESID}$ among species as a function of aridity, MAT, precipitation seasonality, tree cover, maximum wind gust speed, and burned area fraction using multiple regression. Models also included an interaction term between aridity and MAT to test whether the effects of low water availability on tree height, crown size, and shape would be strongest in hotter environments[65], as well as a binary variable testing for systematic differences in $H_{RESID}$, $CD_{RESID}$, and $CAR_{RESID}$ between angiosperms and gymnosperms[9,33,38].

To determine whether $H_{RESID}$, $CD_{RESID}$ and $CAR_{RESID}$ also vary in relation to species' functional traits, we then fit separate models in which either wood density, leaf nitrogen content, SLA or seed mass was added to the multiple regression alongside the environmental predictors described above. Note that models including functional traits as predictors were restricted to the subset of species for which trait data were available (see Supplementary Table 1 for details).

Prior to model fitting, both aridity and seed mass were log-transformed to linearise relationships between response and predictor variables. All continuous predictor variables were then centred and scaled by subtracting the mean and dividing by 1 standard deviation, while the binary variable grouping species into major evolutionary clades was coded as −1 for gymnosperms and 1 for angiosperms. This allowed us to directly compare the effect sizes of different predictors both within and across models based on their regression coefficients. To ensure model coefficients were not affected by collinearity among predictors, we calculated variance inflation factors for all models to confirm they were all ≤ 2.

To account for non-independence among species due to shared evolutionary history, regression models were fit using phylogenetic

generalised least squares (PGLS)[112]. PGLS models were fit using the *gls* function in the *nlme* package[113], where the correlation structure among species was captured using Pagel's λ as implemented by the *corPagel* function in the *ape* package[114]. A phylogenetic tree capturing evolutionary relationships among species was generated using the *V.PhyloMaker* package[115], which uses a comprehensive time-calibrated phylogeny of 79,881 seed plant species as a backbone[116]. Model $R^2$ values that account for the phylogenetic structure of the data were calculated using the *rr2* package[117].

## Reporting summary

Further information on research design is available in the Nature Portfolio Reporting Summary linked to this article.

## Data availability

Data supporting the results of this study are publicly archived on Zenodo (https://doi.org/10.5281/zenodo.14217401). Allometry data from the Tallo database can be accessed here: https://zenodo.org/records/6637599. ICP Forests allometry data are archived here: http://icp-forests.net/page/data-requests. Alberta PSP allometry data are archived here: https://www.alberta.ca/permanent-sample-plots-program.aspx.

## Code availability

R code to replicate the results of this study is publicly archived on Zenodo (https://doi.org/10.5281/zenodo.14217401).

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

## Acknowledgements

We are indebted to the countless researchers and field assistants who helped collect the field data that underpin this study, and without whom this work would not have been possible. TJ was supported by a UK NERC Independent Research Fellowship (grant: NE/S01537X/1), a UKRI Frontier Research grant (grant: EP/Y003810/1) and by a Research Project Grant from the Leverhulme Trust (grant: RPG-2020-341) that also supported FJF. JeC acknowledges an 'Investissement d'Avenir' grant managed by the Agence Nationale de la Recherche (CEBA grant: ANR-10-LABX-25-01 and TULIP grant: ANR-10-LABX-0041). AA was supported by Hebei University (grant: 521100221033), by the Jiangsu Science and Technology Special Project (grant: BX2019084), and Metasequoia Faculty Research Startup Funding at Nanjing Forestry University (grant: 163010230). GJLP was supported by projects DynAfFor (grant: CZZ1636.01D) and P3FAC (grant: CZZ1636.02D) and by the International Foundation for Science (grant: D/5822-1). TRF was funded by NERC (grant: NE/N011570/1). LFA was supported by CAPES and ABC-CNPq (grant: 004/96). LFB was supported by a NERC PhD studentship, RGS-IBG Henrietta Hutton Research Grant, and Royal Society Dudley Stamp Award. BBL was supported by COMPASS-FME, a multi-institutional project supported by the US Department of Energy, Office of Science, Biological and Environmental Research as part of the Environmental System Science Programme. MvB acknowledges funding from the Agua Salud Project, a collaboration between the Smithsonian Tropical Research Institute (STRI), the Panama Canal Authority (ACP) and the Ministry of the Environment of Panama (MiAmbiente), the Smithsonian Institution Forest Global Earth Observatory (ForestGEO), Heising-Simons Foundation, HSBC Climate Partnership, Stanley Motta, Small World Institute Fund, Frank and Kristin Levinson, the Hoch family, the U Trust, the Working Land and Seascapes Programme of the Smithsonian, the National Science Foundation (grant: EAR-1360391), Singapore's Ministry of Education and Yale–NUS College (grant: IG16-LR004). JD and HL were supported by the National Natural Science Foundation of China (grants: 41790422 and 42161144008). BDK was supported by a Western Carolina University Provost Scholarship Development Grant. KD was supported by the African Forest Forum, and the DAAD within the framework of ClimapAfrica (Climate Research for Alumni and Postdocs in Africa) with funds of the Federal Ministry of Education and Research of Germany (grant: 91785431). HG was supported by a NASA GEDI Science Team

Grant (grant: 80NSSC21K0201). YI was supported by Grant-in-Aid for JSPS Fellows (grant: 09J04545). ERL was supported by a UKRI Future Leaders Fellowship (grant: MR/T019832/1). PM acknowledges funding from Alex Fraser Research Forest, Forest Renewal British Columbia, and the Government of British Columbia. EM was supported by the Swedish Energy Agency (grant: 35586-1). JAM was supported by Consejo Nacional de Ciencia y Tecnología (CONACYT; grant: CB-2009-01-128136) and Universidad Nacional Autónoma de México (DGAPA-PAPIIT; grants: IN218416 and IN217620). JMDR and JLHS acknowledge funding from Reinforcing REDD+ and the South-South Cooperation Project, ONAFOR, and USFS. SCR acknowledges funding from FAPEMIG (grant: CAG2327-07), DAAD/CAPE,S and CNPq. GS acknowledges funding by Manchester Metropolitan University's Environmental Science Research Centre. JCD acknowledges funding from the US Department of Energy (grant: DE-SC0023309) and ANR PHYDRAUCC (grant: ANR-ANR-21-CE02-0033-02). MS was funded by a grant from the Ministry of Education, Youth and Sports of the Czech Republic (grant: INTER-TRANSFER LTT19018). HP was supported by the Iranian National Science Foundation (INSF). AS was supported by the Saxon State Ministry for Science, Culture and Tourism (grant: 3-7304/35/6-2021/48880). KT was supported by projects with the National Institute for Environmental Studies and Hokkaido Electric Power Co., Inc., and by KAKENHI, MEXT. ATT acknowledges funding from the NSF (grant: 2003205). MAZ thanks the MAPA-Spain for granting access to the Spanish Forest Inventory data. RM and FB were funded by the Duch Research Council (NOW; grant: ALWOP.457). The study includes data collected by partners of the official UNECE ICP Forests network (http://icp-forests.net/contributors), part of which was co-financed by the European Commission. We thank the Alberta's Ministry of Agriculture, Forestry and Rural Economic Development for access to the Albert PSP allometric data. We are indebted to the countless researchers and field assistants who helped collect the field data compiled in the Tallo database, and without whom this work would not have been possible. Dr Abd Rahman Kassim and Dr Sylvie Gourlet-Fleury, who contributed data to this project, sadly passed away before this paper was completed.

## Author contributions

T.J. conceived the idea for the study. T.J. led the aggregation of the data with assistance from Je.C., D.A.C., Jo.C., A.A., G.J.L.P., T.R.F., D.F., V.A.U. and T.D.J., while S.A.B., L.F.A., M.A., B.A.I., N.P.R.A., C.A., Y.A., N.A., L.F.B., N.B., J.J.B., H.B., Y.E.B., B.B.L., F.B., S.B., Mv.B., A.C., R.C., J.D., M.D., K.D., J.C.D., J.L.D., J.M.D.R., R.A.D., M.E., K.Yv.E., W.F.R., A.F., M.F., E.F., D.I.F., H.G., J.L.G., M.H., J.S.H., J.K.H., A.H., J.L.H.S., S.I.H., R.J.H., K.H., L.B.H., T.I., Y.I., H.S.J., P.R.J., H.K., M.K.L., T.K., B.D.K., T.S.K., Su.K., Sh.K., J.K., S.L., E.R.L., H.L., C.L., J.J.L., Y.M., P.L.M., E.M., R.M., J.A.M., S.M., X.M., S.T.M., G.R.M., F.M., R.M., S.P.N., Z.S.N.H., K.L.O., S.P., R.P., P.L.P., P.P., L.P., M.J.P., H.P., S.C.R., C.R., An.S., J.S., Mi.S., G.S., A.l.S., B.S., F.J.S., Ma.S., K.T., A.T.T., M.A.V., A.V., M.C.V., A.G.V., P.W., W.W., L.Q.W., C.W., M.W., W.X., Fd.A.X., Y.X., T.Y., M.A.Z. and N.E.Z. all contributed data. T.J. performed the analyses with the assistance of F.J.F. and input from all co-authors. T.J. wrote the first draft of the manuscript, with all co-authors providing editorial input.

## Competing interests

The authors declare no competing interests.

## Additional information

**Tommaso Jucker** [ID][1] ✉, **Fabian Jörg Fischer**[1], **Jérôme Chave** [ID][2], **David A. Coomes** [ID][3], **John Caspersen**[4], **Arshad Ali** [ID][5], **Grace Jopaul Loubota Panzou**[6,7], **Ted R. Feldpausch** [ID][8], **Daniel Falster** [ID][9], **Vladimir A. Usoltsev**[10], **Toby D. Jackson** [ID][1], **Stephen Adu-Bredu**[11], **Luciana F. Alves** [ID][12], **Mohammad Aminpour**[13], **Bhely Angoboy Ilondea**[14,15], **Niels P. R. Anten**[16], **Cécile Antin**[17], **Yousef Askari**[18], **Narayanan Ayyappan** [ID][19], **Lindsay F. Banin** [ID][20], **Nicolas Barbier** [ID][17], **John J. Battles**[21], **Hans Beeckman** [ID][22], **Yannick E. Bocko**[7], **Ben Bond-Lamberty** [ID][23], **Frans Bongers** [ID][24], **Samuel Bowers**[25], **Michiel van Breugel** [ID][26,27,28], **Arthur Chantrain**[29], **Rajeev Chaudhary**[30], **Jingyu Dai**[31], **Michele Dalponte**[32], **Kangbéni Dimobe**[33], **Jean-Christophe Domec**[34,35], **Jean-Louis Doucet**[29], **Juan Manuel Dupuy Rada**[36], **Remko A. Duursma**[37], **Moisés Enríquez** [ID][38], **Karin Y. van Ewijk**[39], **William Farfán-Rios** [ID][40], **Adeline Fayolle**[29,41], **Marco Ferretti** [ID][42], **Eric Forni**[41], **David I. Forrester** [ID][43], **Hammad Gilani**[44], **John L. Godlee** [ID][25], **Matthias Haeni** [ID][42], **Jefferson S. Hall**[27], **Jie-Kun He** [ID][45], **Andreas Hemp** [ID][46], **José L. Hernández-Stefanoni** [ID][36], **Steven I. Higgins**[47], **Robert J. Holdaway**[48], **Kiramat Hussain**[49], **Lindsay B. Hutley** [ID][50], **Tomoaki Ichie**[51], **Yoshiko Iida** [ID][52], **Hai-Sheng Jiang** [ID][45], **Puspa Raj Joshi**[53], **Hasan Kaboli**[54], **Maryam Kazempour Larsary** [ID][55], **Tanaka Kenzo**[56], **Brian D. Kloeppel** [ID][57], **Takashi S. Kohyama**[58], **Suwash Kunwar**[30,59], **Shem Kuyah** [ID][60], **Jakub Kvasnica** [ID][61], **Siliang Lin**[62], **Emily R. Lines** [ID][63],

Hongyan Liu [31], Craig Lorimer [64], Jean-Joël Loumeto [7], Yadvinder Malhi [65], Peter L. Marshall [66], Eskil Mattsson [67,68], Radim Matula [69], Jorge A. Meave [38], Sylvanus Mensah [70,71], Xiangcheng Mi [72], Stéphane T. Momo [17,73], Glenn R. Moncrieff [74,75], Francisco Mora [76], Rodrigo Muñoz [24,38,77], Sarath P. Nissanka [78], Zamah Shari Nur Hajar [79], Kevin L. O'Hara [21], Steven Pearce [80], Raphaël Pelissier [17], Pablo L. Peri [81], Pierre Ploton [17], Lourens Poorter [24], Mohsen Javanmiri Pour [82], Hassan Pourbabaei [55], Sabina C. Ribeiro [83], Casey Ryan [25], Anvar Sanaei [84], Jennifer Sanger [80], Michael Schlund [85], Giacomo Sellan [86,87], Alexander Shenkin [88], Bonaventure Sonké [73], Frank J. Sterck [24], Martin Svátek [61], Kentaro Takagi [89], Anna T. Trugman [90], Matthew A. Vadeboncoeur [91], Ahmad Valipour [92], Mark C. Vanderwel [93], Alejandra G. Vovides [94,95], Peter Waldner [42], Weiwei Wang [72], Li-Qiu Wang [59], Christian Wirth [84,96], Murray Woods [97], Wenhua Xiang [98], Fabiano de Aquino Ximenes [99], Yaozhan Xu [100,101], Toshihiro Yamada [102], Miguel A. Zavala [103] & Niklaus E. Zimmermann [42]

[1]School of Biological Sciences, University of Bristol, Bristol BS8 1TQ, UK. [2]UMR5300 Centre de Recherche sur la Biodiversité et l'Environnement, CNRS, INPT, IRD, Université de Toulouse, Bât 4R1, 118 route de Narbonne, 31062 Toulouse, France. [3]Conservation Research Institute, University of Cambridge, Cambridge CB2 3EA, UK. [4]Institute of Forestry and Conservation, University of Toronto, 33 Willcocks Street, Toronto, ON M5S 3B3, Canada. [5]Forest Ecology Research Group, College of Life Sciences, Hebei University, Baoding 071002 Hebei, China. [6]Institut Supérieur des Sciences Géographiques, Environnementales et de l'Aménagement (ISSGEA), Université DENIS SASSOU-N'GUESSO, Kintélé, République du Congo. [7]Laboratoire de Biodiversité, de Gestion des Ecosystèmes et de l'Environnement (LBGE), Faculté des Sciences et Techniques, Université Marien NGOUABI, BP 69 Brazzaville, Brazzaville, République du Congo. [8]Faculty of Environment, Science and Economy, University of Exeter, Exeter EX4 4QE, UK. [9]Evolution & Ecology Research Centre, University of New South Wales Sydney, NSW Sydney, Australia. [10]Department of Forest Mensuration and Management, Ural State Forest Engineering and Economic University, Yekaterinburg, Russia. [11]Forestry Research Institute of Ghana, Council for Scientific and Industrial Research, University, Kumasi, Ghana. [12]Center for Tropical Research, Institute of the Environment and Sustainability, University of California Los Angeles, Los Angeles, CA, USA. [13]Natural Recourses and Watershed Management Office, West Azerbaijan Province, Urmia, Iran. [14]Institut National pour l'Etude et la Recherche Agronomiques, BP 2037 Kinshasa, Democratic Republic of the Congo. [15]Université Pédagogique Nationale, BP 8815 Kinshasa-Ngaliema, Democratic Republic of the Congo. [16]Center for Crop Systems Analysis, Wageningen University & Research, Wageningen, Netherlands. [17]AMAP lab, Montpellier University, IRD, CIRAD, CNRS, INRAE, Montpellier, France. [18]Research Division of Natural Resources, Kohgiluyeh and Boyerahmad Agriculture and Natural Resources Research and Education Center, AREEO, Yasouj, Iran. [19]Department of Ecology, French Institute of Pondicherry, Puducherry 605014, India. [20]UK Centre for Ecology & Hydrology, Edinburgh, UK. [21]University of California Berkeley, Berkeley, CA 94720, USA. [22]Service of Wood Biology, Royal Museum for Central Africa, Tervuren, Belgium. [23]Joint Global Change Research Institute, Pacific Northwest National Laboratory, 5825 University Research Ct. #3500, College Park, MD 20740, USA. [24]Forest Ecology and Forest Management Group, Wageningen University & Research, P.O. Box 47, 6700 AA Wageningen, Netherlands. [25]School of GeoSciences, University of Edinburgh, Edinburgh EH9 3FF, UK. [26]Yale-NUS College, 12 College Avenue West, 138610 Singapore, Singapore. [27]ForestGEO, Smithsonian Tropical Research Institute, Apartado Postal 0843-03092, Panama, Republic of Panama. [28]Department of Geography, National University of Singapore, 1 Arts Link, #03-01 Block AS2, 117570 Singapore, Singapore. [29]Université de Liège, Gembloux Agro-Bio Tech, Gembloux, Belgium. [30]Division Forest Office, Ministry of Forest, Sudurpashchim province, Dhangadhi, Nepal. [31]College of Urban and Environmental Sciences and MOE Laboratory for Earth Surface Processes, Peking University, Beijing 100871, China. [32]Research and Innovation Centre, Fondazione Edmund Mach, via E. Mach 1, 38098 San Michele all'Adige, TN, Italy. [33]Département des Eaux, Forêts et Environnement, Institut des Sciences de l'Environnement et du Développement Rural, Université Daniel Ouezzin Coulibaly, BP 176 Dédougou, Burkina Faso. [34]Bordeaux Sciences Agro-UMR ISPA, INRAE, Bordeaux, France. [35]Nicholas School of the Environment, Duke University, Durham, NC, USA. [36]Centro de Investigación Científica de Yucatán A.C. Unidad de Recursos Naturales, Calle 43 #130, Colonia Chuburná de Hidalgo, C.P, 97205 Mérida, Yucatán, México. [37]Statistics Netherlands, Henri Faasdreef 312, 2492 JP Den Haag, Netherlands. [38]Departamento de Ecología y Recursos Naturales, Facultad de Ciencias, Universidad Nacional Autónoma de México. Coyoacán, Ciudad de México C.P 04510, Mexico. [39]Department of Geography and Planning, Queen's University, Kingston, ON, Canada. [40]Department of Biology, Washington University in St Louis, St Louis, MO 63130, USA. [41]CIRAD, UPR Forêts et Sociétés, F-34398 Montpellier, France. [42]Swiss Federal Research Institute for Forest, Snow and Landscape Research WSL, Zürcherstrasse 111, CH-8903 Birmensdorf, Switzerland. [43]CSIRO Environment, GPO Box 1700, Canberra, ACT, Australia. [44]Plant and Environmental Sciences, New Mexico State University, Las Cruces, NM 88003, USA. [45]Spatial Ecology Lab, School of Life Sciences, South China Normal University, Guangzhou, 510631 Guangdong, China. [46]University of Bayreuth, Department of Plant Systematics, Universitätsstr. 30-31, 95440 Bayreuth, Germany. [47]Department of Botany, University of Otago, PO Box 56 Dunedin 9016, New Zealand. [48]Landcare Research, PO Box 69040 Lincoln 7640, New Zealand. [49]Gilgit-Baltistan Forest Wildlife and Environment Department, Gilgit, Pakistan. [50]Research Institute for the Environment & Livelihoods, Charles Darwin University, Northern Territory, Casuarina, NSW, Australia. [51]Faculty of Agriculture and Marine Science, Kochi University, B200 Monobe, Nankoku, Kochi 783-8502, Japan. [52]Forestry and Forest Products Research Institute, 1 Matsunosato, Tsukuba, Ibaraki 305-8687, Japan. [53]Institute of Forestry, Tribhuvan University, Hetauda Campus, Hetauda 44107, Nepal. [54]Faculty of Desert Studies, Semnan University, Semnan, Iran. [55]Department of Forestry, Faculty of Natural Resources, University of Guilan, Somehsara 43619-96196, Iran. [56]Japan International Research Center for Agricultural Sciences, Tsukuba, Ibaraki 305-8686, Japan. [57]Office of the Vice Provost for International Affairs, Princeton University, Princeton, NJ 08544, USA. [58]Faculty of Environmental Earth Science, Hokkaido University, Sapporo 060-0810, Japan. [59]Department of Forest Resources Management, College of Forestry, Nanjing Forestry University, Nanjing 210037 Jiangsu, China. [60]Jomo Kenyatta University of Agriculture and Technology (JKUAT), 62000, 00200 Nairobi, Kenya. [61]Department of Forest Botany, Dendrology and Geobiocoenology, Faculty of Forestry and Wood Technology, Mendel University in Brno, Brno, Czech Republic. [62]Guangdong Provincial Key Laboratory of High Technology for Plant Protection, Plant Protection Research Institute, Guangdong Academy of Agricultural Sciences, Guangzhou, 510640 Guangdong, China. [63]Department of Geography, University of Cambridge, Downing Place, Cambridge CB2 3EN, UK. [64]Department of Forest and Wildlife Ecology, University of Wisconsin-Madison, Madison, WI 53706, USA. [65]Environmental Change Institute, School of Geography and the Environment, University of Oxford, Oxford, UK. [66]Faculty of Forestry, University of British Columbia, Vancouver, BC, Canada. [67]IVL Swedish Environmental Research Institute, Aschebergsgatan 44, 411 33 Göteborg, Sweden. [68]Gothenburg Global Biodiversity Centre (GGBC), Gothenburg, Sweden. [69]Faculty of Forestry and Wood Sciences, Czech University of Life Sciences Prague, Prague 6, Suchdol, Czech Republic. [70]Laboratoire de Biomathématiques et d'Estimations Forestières, Faculté des Sciences Agronomiques, Université d'Abomey Calavi, Cotonou, Benin. [71]Chair of Forest Growth and Dendroecology, Albert-Ludwigs-Universität Freiburg, 79106 Freiburg, Germany. [72]State Key Laboratory of Vegetation and Environmental Change, Institute of Botany, Chinese Academy of Sciences, Beijing 100093,

China. [73]Laboratoire de Botanique systématique et d'Ecologie, Département des Sciences Biologiques, Ecole Normale Supérieure, Université de Yaoundé I, Yaoundé, Cameroon. [74]Global Science, The Nature Conservancy, Cape Town, South Africa. [75]Centre for Statistics in Ecology, Environment and Conservation, Department of Statistical Sciences, University of Cape Town, Private Bag X3, Rondebosch 7701, South Africa. [76]Instituto de Investigaciones en Ecosistemas y Sustentabilidad, Universidad Nacional Autónoma de México, Morelia, Michoacán, Mexico. [77]Wageningen Environmental Research, Wageningen University & Research, 6708PB Wageningen, Netherlands. [78]Department of Crop Science, Faculty of Agriculture, University of Peradeniya, Peradeniya, Sri Lanka. [79]Forestry and Environment Division, Forest Research Institute Malaysia, Kepong, Selangor 52109, Malaysia. [80]The Tree Projects, Hobart, TAS, Australia. [81]Universidad Nacional de la Patagonia Austral (UNPA) - Instituto Nacional de Tecnología Agropecuaria (INTA) - CONICET, CC 332, (9400), Río Gallegos, Santa Cruz, Argentina. [82]Agriculture and Natural Resources Research and Education Center, Kermanshah Province, Agricultural Research, Extension and Education Organization, Kermanshah, Iran. [83]Centro de Ciências Biológicas e da Natureza, Universidade Federal do Acre, Campus Universitário, BR 364, Km 04, Distrito Industrial, Rio Branco, Acre 69920-900, Brazil. [84]Systematic Botany and Functional Biodiversity, Institute of Biology, Leipzig University, Leipzig, Germany. [85]Department of Natural Resources, Faculty of Geo-information Science and Earth Observation (ITC), University of Twente, Hengelosestraat 99, Enschede 7514AE, Netherlands. [86]UMR EcoFoG, CIRAD, Campus Agronomique, 97310 Kourou, French Guiana. [87]Department of Natural Sciences, Manchester Metropolitan University, Chester Street, Manchester M1 5GD, UK. [88]School of Informatics, Computing, and Cyber Systems, Northern Arizona University Flagstaff, Flagstaff, AZ, USA. [89]Field Science Center for Northern Biosphere, Hokkaido University, Horonobe 098-2943, Japan. [90]Department of Geography, University of California Santa Barbara, Santa Barbara, CA 93106, USA. [91]Earth Systems Research Center, University of New Hampshire, Durham, NH 03824, USA. [92]Department of Forestry and Dr. Hedayat Ghazanfari Center for Research and Development of Northern Zagros Forestry, University of Kurdistan, Kurdistan, Iran. [93]Department of Biology, University of Regina, 3737 Wascana Pkwy, Regina, SK S4S 0A2, Canada. [94]Institute of Biology and Environmental Sciences, Carl von Ossietzky University of Oldenburg, Oldenburg 26129, Germany. [95]School of Geographical and Earth Sciences, University of Glasgow, East Quadrangle, Glasgow, UK. [96]German Centre for Integrative Biodiversity Research (iDiv) Halle-Jena-Leipzig, Leipzig, Germany. [97]Ontario Ministry of Natural Resources, North Bay, ON P1A 4L7, Canada. [98]Faculty of Life Science and Technology, Central South University of Forestry and Technology, Changsha 410004 Hunan, China. [99]Forest Science, New South Wales Department of Primary Industries and Regional Development, Locked Bag 5022, Parramatta, NSW 2124, Australia. [100]State Key Laboratory of Aquatic Botany and Watershed Ecology, Wuhan Botanical Garden, Chinese Academy of Sciences, Wuhan 430074, China. [101]Center of Conservation Biology, Core Botanical Gardens, Chinese Academy of Sciences, Wuhan 430074, China. [102]Graduate School of Integrated Sciences of Life, Hiroshima University, Hiroshima 739-8521, Japan. [103]Universidad de Alcalá, Forest Ecology and Restoration Group (FORECO), Departamento de Ciencias de la Vida, 28805 Alcalá de Henares, Madrid, Spain. ✉e-mail: t.jucker@bristol.ac.uk

