## [Transparent Peer Review file · Nature Communications]

The global spectrum of tree crown architecture

Corresponding Author: Dr Tommaso Jucker

Version 0:

Reviewer comments:

Reviewer #1

(Remarks to the Author)

The authors provide a robust and extensive analysis of the global distribution of 3 key structural traits in trees by assembling a massive database of individual trait measurements. They use this dataset to test the sensitivity of traits to environmental factors and search for evidence of phylogenetic signatures. Overall, the work is very thorough and brings together years of work from the field in a novel and informative way, while providing support for many generalized hypothesized trait sensitivities in a wide range of species. This is the type of information that could greatly improve a range of tree growth models, vegetation models, remote sensing, and radiative transfer models moving forward. These very same approaches are being applied (as alluded to in the discussion) in the Global TLS Database (global-tls.net) and this creates a nice structure for similar analyses of 3D tree traits.

The work is novel and builds on decades of field measurements across most major forest biomes. The work is an excellent contribution to the field of macro ecology and will help inform aspects of plant scaling relationships and sensitivities to climate.

In general, the authors are careful to not draw definitive conclusions about the generalized trends they found in the data, but rather offer substantial evidence for each claim to support the major observations regarding plant trait sensitivity. The one area of the work that probably needs a bit more discussion is the role of uncertainty and variation in the findings. Some of the analyses do not explicitly incorporate uncertainty or variation in the estimates. Since the approaches here rely on averaged remotely sensed climate estimates, we can expect there is a bit more uncertainty in the final estimates than reported. That said, I think the authors are taking a reasonable approach and adding propagated and quantified uncertainty estimates are unlikely to change the final findings.

I see no issues regarding data analysis, interpretation or conclusions. In general, the authors had a robust analysis approach that provided insight on the specific hypotheses tested. As stated above, a structured approach to uncertainty propagation would be appreciated, but would be unlikely to change the overall findings of the study.

The methodology is sound and the approaches are in line with expectations for this type of study. My one suggestion would be for the authors to incorporate uncertainty in the analyses more explicitly. For instance, (if I'm not mistaken) the mean of specific climate/ecosystem variables was used for entire species, when these species may exist across a range of growing conditions. In other words, variance exists that may be important in species that are generalists and exist in many places.

Enough detail is provided for the methods to be reproduced and code for the analysis is provided. I hope the authors will publish the final cleaned and assembled dataset for others to use.

The following line-by-line comments were noted while reviewing:

Line 223: A nice, simple introduction that is concrete.

Line 231: Succinct explanation that covers the main applications to remote sensing and vegetation models. Great.

Line 243: Solid summary of the main research on this topic.

Line 266: Excellent sample size. The presented figure of the distribution of sample locations should be colored to show number of samples per aggregate unit. This would be a more transparent way of showing the distribution of individual trees in the dataset. It would also be useful to show the aggregate number of samples as a distribution over the latitude and longitude of the global map. The second suggestion is less essential, but would provide some more understanding of the data distribution than the presence / absence map as it is shown now.

Lines: 275-284 Will this compiled and standardized database be made available somewhere? The location should be cited.

Lines 299-311 I appreciate taking different analysis approaches, but this first one seems a bit arbitrary and it's unclear if it necessarily adds anything useful (compared to the second approach) to the analysis. Maybe introducing this section with a justification for why this simpler approach is used.

Lines: 360 Please provide detail on why 2008 was chosen as the baseline year.

Line 373: For each of these species have you reported the mean and standard deviation of your estimates (e.g. mean and standard deviation of precipitation for a single species, etc.)? If not please report them in a table in the supplement. This will provide an estimate of the uncertainty / variation across species (especially those species that exist in many different growing environments).

Line 1104: It would be helpful if the colors were ordered as shown in the legend on the top of the figure. For instance, group all purple, grey, and yellow/red into a block. The colors and orders of colors are seemingly grouped by the "low, med, high" groupings, but I'm not sure if this is easiest for interpretation of the proportions of each class in each biome.

Looking forward to seeing the published version of this work!

Atticus Stovall

(Remarks on code availability)

I was able to download and run the majority of the code provided. No README was included but the comments were helpful for understanding the different analysis sections. I was unable to evaluate any of the tree-level analysis because these data were not provided, but I imagine the dataset is quite large.

Reviewer #2

(Remarks to the Author)

This is a timely article and it seeks to answer urgent questions on tree allometry. However, authors need to clarify the data analysis.

-What are the noteworthy results?

Tree heights were constrained by light competition and water availability while exposure to wind and fire constrained crown size.

- Will the work be of significance to the field and related fields? How does it compare to the established literature? If the work is not original, please provide relevant references.

Yes. It is unclear whether reported scaling exponents conformed to theoretical predictions. Given the large dataset, it will be interesting to investigate if scaling exponents of the various allometries tested are in line with existing theoretical predictions such as the stress similarity model, metabolic scaling theory, geometric similarity model.

- Does the work support the conclusions and claims, or is additional evidence needed?

Yes once the corrections are made

- Are there any flaws in the data analysis, interpretation and conclusions? - Do these prohibit publication or require revision?

Maybe. Allometry parameters can widely vary with regression techniques. What was the rationale for using log-transformed data? Secondly, selection of allometric models follows a common scientific procedure where different models are experimented and tested, and various indices/coefficient (such as R, adjustedR², RMSE, AIC or BIC, etc.) used in selecting the model that best fits or explains variances in the data. I am seriously questioning how the authors arrived at opting for the log-log regression model. If this model performance was better than other models considered in your analysis? What other models were considered and assessed? These details are necessary to understand the analysis and information reported. I am questioning whether there might be other models which could have fitted the data better than the log-log regression model?

What are the merits of the transformed log-log model? Why did it perform better in explaining variance in the data used for your analyses? I am curious if the log-log model best explained the allometry of all the tree species in this study? Can you provide supporting information or discussions to clarify this?

In general, SMA estimates of lines summarizing relationship between two variables are superior to ordinary least square

linear regression and have been advocated in the literature because residual variance is minimized in both the X and Y dimensions rather than in the Y dimension only. When comparing regressions, differences can occur in either exponent of a (Y intercept) and/or b (regression slope). If b differs among species, species with larger b will have greater increase in Y per increment of X. If a differs, but b does not, species with larger a will have a consistently larger amount of Y at any given value of X (Kohyama and Hotta, 1990; Sposito and Santos, 2001). Authors should clarify how the ordinary least square regression was suitable for the analysis.

- Is the methodology sound? Does the work meet the expected standards in your field?

Yes

- Is there enough detail provided in the methods for the work to be reproduced?

yes

(Remarks on code availability)

Reviewer #3

(Remarks to the Author)

Review of The global spectrum of tree architecture

Jucker et al. provide and analyze a prodigious dataset on the global relationship of tree height (H), crown diameter (D), and crown aspect ratio (D/H) to climate, disturbance, competition, other functional traits, phylogeny and each other. Height and height scaling relationships appear to be most strongly affected by water availability and competition; crown width, by exposure to disturbance by wind or fire. Some lineages had unusual crown forms apparently tied more to phylogeny than environment.

This is an important paper that quantifies and crystallizes several patterns evident, in many cases, to natural historians for decades if not centuries. The global nature of the database, the extraordinary range of plots, species, and evolutionary lineages assessed, the sound statistical frameworks (in most cases) employed to analyze the data, and the conclusions reached will likely make this paper a touchstone for years to come. The authors are to be congratulated.

However, as currently written, the manuscript has two central, interrelated flaws: a lack of a conceptual framework to predict or account for patterns in tree height, diameter, and crown aspect ratio, and a failure to cite some centrally important papers that provide such a framework. These flaws must be addressed.

Specifically, the authors should first cite Givnish et al. 2015 (*Ecology* 95: 2991-3007, Determinants of maximum tree height in *Eucalyptus* species along a rainfall gradient in Victoria, Australia) AND briefly summarize their general model relating maximum tree height to climate, soil fertility, and disturbance, to the allometry of allocation to wood vs. leaves with height, and to vertical stratification of environmental conditions with height above the ground. That paper shows a strong relationship of tree height to climatic moisture supply relative to demand, as quantified by P/Ep (mean annual precipitation divided by pan evaporation) – which is closely related to the aridity index used by Jucker et al. – and argues for its generality, albeit modulated by differences in temperature, latitude, soil fertility, and disturbance. These theoretical and empirical findings are directly relevant to the subject of the current paper, and they really must be cited – not as a contentless superscript, but for specific predictions and statistical relationships obtained along a substantial climatic gradient. Those earlier findings are directly relevant to what is arguably the most important conclusions of the current paper. • Given that the authors tie global variation in height and crown aspect ratio (tied to height) to leaf nitrogen content and specific leaf area, they should also cite Givnish et al. 2015 for both their model predictions and findings of such relationships in their *Eucalyptus* study ... and for the climatic and edaphic drivers of variation in leaf N and SLA. They might also consider citing Smith et al. 2023 (*Nature Communications* 14: 7173, Ecophysiological adaptations shape distributions of closely related trees along a climatic moisture gradient) that detail shifts in 50 functional traits, as well as height growth, mass growth, and survival in 10 species of *Eucalyptus* grown in common gardens along the same gradient. That paper shows that species differences in many of those traits are heritable and largely accord with economic models to maximize rates of height growth.

Second, the authors should cite King 1981 (*Oecologia* 51: 351-356, Tree dimensions: Maximizing the rate of height growth in dense stands) and Givnish 1995 (*Plant stems: biomechanical adaptations for energy capture and influence on species distributions*. Pp. 3-49 in B. L. Gartner (ed.), *Plant Stems: Physiology and Functional Morphology*. Chapman and Hall, New York). These papers explain why the ratio of tree crown width to height should converge on a specific value in crowded stands. The first paper presents a quantitative model for the optimal allocation of energy between crown and bole; the second shows that the optimal value of that ratio corresponds to crown width increasing in proportion to crown height (i.e., crown width/height ratio is constant) and accounts for the observed relationship in fully stocked (crowded) stands documented by O'Neill & DeAngelis 1981 (Pp. 411-449 in DE Reichle (ed), *Dynamic Properties of Forest Ecosystems*, Cambridge University Press) and used by Givnish 1986 (*Journal of Theoretical Biology* 119: 139-146, Biomechanical constraints on self-thinning in plant populations) to provide the first mechanistic explanation for the -3/2 power thinning law in plants. Givnish 1995 provides an extensive table of predictions for how difference aspects of crown form should vary with environmental conditions. Arguments for broader crowns relative to height in shade-adapted trees are presented, as well as supporting data from an extensive dataset on North American champion trees.

Third, while the authors' statistical approaches are generally sound, I note two weaknesses that they should either justify or redress: (A) In studies involving allometric variation (e.g., between crown height and width), Type I regression models that

minimize the residual variance of the dependent variable based on relationships to independent variables – i.e., ordinary linear regressions, as employed here) are often viewed as inappropriate, given that there is no clearly distinction between dependent and independent variables. Type II regressions, which minimize the residual variance about the regression in both the x- and y-directions, have generally been employed in such cases. The authors should justify their use of Type I regression or replace it with Type II regression. I am not sure whether a Type II version of generalized linear models is available however, and so they may have to use the current version of GLM for those (important) analyses. (B) It's generally recognized that phylogenetic regression – which accounts for relationships among species, and thus the non-independence of data points – should be used in comparative studies like those at the heart of this paper. I recommend that the authors should at least present phylogenetic regressions that complement their traditional regressions, even if they rely mainly on the latter. They already have relationships in hand for many of their study species, as shown by Figure 3.

Fourth, I strongly recommend that the authors revise Figure 3 so as to increase its impact, by (A) stretching it vertically a bit, to give better resolution of individual branches, and then (B) using maximum likelihood mapping to colorize branches to visualize the evolution of tree height, crown width, and crown aspect ratio. Widely available R packages by Liam Revell can be used for this task.

Thomas J. Givnish
Henry Allan Gleason Professor of Botany
Wilhelm Hofmeister Professor of Botany
University of Wisconsin-Madison

(Remarks on code availability)

Version 1:

Reviewer comments:

Reviewer #1

(Remarks to the Author)

The revisions in the current version of the paper addressed all of my previous comments. I am satisfied with the current version and believe it is ready for publication. Nice work.

(Remarks on code availability)

Reviewer #2

(Remarks to the Author)

The authors did a fantastic job clarifying their analysis and methods and providing supplementary files. I recommend it should be accepted.

(Remarks on code availability)

Reviewer #3

(Remarks to the Author)

Jucker et al. have responded adequately to most of the reviewers' comments. Their efforts are, in general, to be applauded.

HOWEVER, they have failed to respond to my central criticism – that they need to provide a simple conceptual framework for ecological trends in tree height and crown width vs. height. Worse, their rebuttal letter misrepresents what they actually did in editing the manuscript. The manuscript text must be changed. As detailed below, the changes needed are minimal – rewriting three sentences – but crucial.

To quote their letter on two key points:

1. **COMMENT:** Specifically, the authors should first cite Givnish et al. 2015 (*Ecology* 95: 2991-3007, Determinants of maximum tree height in Eucalyptus species along a rainfall gradient in Victoria, Australia) AND briefly summarize their general model relating maximum tree height to climate, soil fertility, and disturbance, to the allometry of allocation to wood vs. leaves with height, and to vertical stratification of environmental conditions with height above the ground. That paper shows a strong relationship of tree height to climatic moisture supply relative to demand, as quantified by P/Ep (mean annual precipitation divided by pan evaporation) – which is closely related to the aridity index used by Jucker et al. – and argues for its generality, albeit modulated by differences in temperature, latitude, soil fertility, and disturbance. These theoretical and empirical findings are directly relevant to the subject of the current paper, and they really must be cited – not

as a content-less superscript, but for specific predictions and statistical relationships obtained along a substantial climatic gradient. Those earlier findings are directly relevant to what is arguably the most important conclusions of the current paper. Given that the authors tie global variation in height and crown aspect ratio (tied to height) to leaf nitrogen content and specific leaf area, they should also cite Givnish et al. 2015 for both their model predictions and findings of such relationships in their Eucalyptus study ... and for the climatic and edaphic drivers of variation in leaf N and SLA. They might also consider citing Smith et al. 2023 (Nature Communications 14: 7173, Ecophysiological adaptations shape distributions of closely related trees along a climatic moisture gradient) that detail shifts in 50 functional traits, as well as height growth, mass growth, and survival in 10 species of Eucalyptus grown in common gardens along the same gradient. That paper shows that species differences in many of those traits are heritable and largely accord with economic models to maximize rates of height growth.

Response: Thank you for this, it is really helpful both in terms of framing the analysis of variation in tree height along aridity gradients, but also relating this back to other functional traits such as leaf nitrogen content. We have now incorporated Givnish et al. 2015 into the text, including highlighting it in the introduction and citing it as a key reference supporting our predictions outlined in Table 1. We have also refer back to it at several points in the discussion, citing it as a key example supporting our findings about how investment in height growth varies along aridity gradients and how we might expect this to covary with investment in leaf nitrogen."

My response: Not only do the authors not "highlight" the conceptual framework of Givnish et al. 2015 in the introduction, they do not cite it all, let alone give a simple sentence summarizing its key conceptual framework re what determines tree height. Furthermore, I fundamentally disagree with their explanation for why height growth along aridity gradients should covary with leaf nitrogen.

This is unacceptable, both for materially ignoring the integrative conceptual framework of Givnish et al., and for stating things in their rebuttal letter that are simply untrue. The authors have stuck to their original approach and not changed their intro in any way (based on the tracked-changes version), except to add a final paragraph about some trends in crown form they document later.

I recommend the authors remedy this situation – and their failure to respond to my second main criticism (see point 2 below) by inserting the following two sentences on line 239:

Maximum tree height should increase toward moister or more fertile sites, based on the balance between photosynthesis vs. increases with height in allocation to unproductive stem tissue and hydraulic limitations to photosynthesis. The ratio of crown diameter to height that maximizes upward growth should not vary with tree height, but relative crown width should be greater in more open habitats (or under closed canopies) where height growth increases energy capture less than horizontal crown expansion^{7,32}.

Reference A: Givnish, T. J., Wong, S. C., Stuart-Williams, H., Holloway-Phillips, M. & Farquhar, G. D. Determinants of maximum tree height in Eucalyptus species along a rainfall gradient in Victoria, Australia. Ecology 95, 2991–3007 (2014).

Rationale: (1) The authors need to cite and highlight the Givnish et al. 2015 framework, based on my earlier criticism and their own statements in the rebuttal letter. (2) Earlier in the intro paragraph, they make the conceptually erroneous statement that "For instance, in arid climates woody biomass allocation tends to shift away from height growth to limit the risk of hydraulic failure, resulting in trees that are shorter for a given diameter." As Givnish et al. 2015 showed, including only hydraulic limitation does not account for shifts in tree height with increasing aridity; the allometric variation in allocation to unproductive stem tissue must also be included. (3) The second recommended sentence addresses my second main criticism (see next paragraph) and provides a pointer to the conceptual framework for relative crown width.

2. Comment: Second, the authors should cite King 1981 (Oecologia 51: 351-356, Tree dimensions: Maximizing the rate of height growth in dense stands) and Givnish 1995 (Plant stems: biomechanical adaptations for energy capture and influence on species distributions. Pp. 3-49 in B. L. Gartner (ed.), Plant Stems: Physiology and Functional Morphology. Chapman and Hall, New York). These papers explain why the ratio of tree crown width to height should converge on a specific value in crowded stands. The first paper presents a quantitative model for the optimal allocation of energy between crown and bole; the second shows that the optimal value of that ratio corresponds to crown width increasing in proportion to crown height (i.e., crown width/height ratio is constant) and accounts for the observed relationship in fully stocked (crowded) stands documented by O'Neill & DeAngelis 1981 (Pp. 411-449 in DE Reichle (ed), Dynamic Properties of Forest Ecosystems, Cambridge University Press) and used by Givnish 1986 (Journal of Theoretical Biology 119: 139-146, Biomechanical constraints on self-thinning in plant populations) to provide the first mechanistic explanation for the -3/2 power thinning law in plants. Givnish 1995 provides an extensive table of predictions for how difference aspects of crown form should vary with environmental conditions. Arguments for broader crowns relative to height in shade-adapted trees are presented, as well as supporting data from an extensive dataset on North American champion trees.

Response: Again, thank you for highlighting this important work. We agree that it's really important to link our finding back to earlier work that set the foundations for our current understanding the biophysical constraints that shape tree architecture. The papers by King 1981 and Givnish 1995 are particularly relevant to the opening two paragraph of the introduction, where we provide a broad overview of the various constraints that trees operate under when expanding their crowns both vertically and horizontally. We have also incorporated them in Table 1, where they help support our hypotheses about how we might expect tree height and crown width to vary in relation to competition for light with neighbours. Finally, they help support

several of our key findings, and we have therefore referred to them specifically in the revised discussion.

My response: Again, the authors did not – so far as I can see – insert any statement addressing the importance of these papers and the ideas and data they present for setting relative crown width. The second sentence I proposed adding (see point 1) would remedy that.

3. Finally, I fundamentally disagree with the authors' explanation for why tree height should increase with leaf nitrogen content (lines 505-508). They ignore the effect of increased photo-synthesis per unit leaf mass (which increases with leaf N content) on maximum tree height. They ignore the effect of lowered allocation to roots in causing greater tree height on moister or more fertile sites AND on increasing whole-plant shade tolerance (see reference 7). They latter undercuts their argument for increased allocation to height growth ... and, of course, fails to account for decreased whole-plant shade tolerance with height (a key point made by reference 7). The authors have no data showing an increased investment in height growth at a given height on sites with higher leaf N content. Citing our paper to implicitly endorse their argument makes me uncomfortable.

To cope with these problems, I recommend simply re-writing the problematic lines 505-508 as follows:

Based on this we would expect species with higher leaf nitrogen to invest more in both height growth and crown expansion to allow them to intercept more light, limit self-shading and optimise the distribution of leaves across their crowns. In addition, higher leaf nitrogen content should lead to greater photosynthetic rates and result in greater maximum tree height – as found previously along a gradient of increasing rainfall in AustraliaA.

A again is Givnish et al. 2015 (see above).

If the authors make the two changes listed above, under points 1 and 3, I would fully support publication of their important manuscript.

(Remarks on code availability)

Reviewer #1

Comment: The authors provide a robust and extensive analysis of the global distribution of 3 key structural traits in trees by assembling a massive database of individual trait measurements. They use this dataset to test the sensitivity of traits to environmental factors and search for evidence of phylogenetic signatures. Overall, the work is very thorough and brings together years of work from the field in a novel and informative way, while providing support for many generalized hypothesized trait sensitivities in a wide range of species. This is the type of information that could greatly improve a range of tree growth models, vegetation models, remote sensing, and radiative transfer models moving forward. These very same approaches are being applied (as alluded to in the discussion) in the Global TLS Database (global-tls.net) and this creates a nice structure for similar analyses of 3D tree traits. The work is novel and builds on decades of field measurements across most major forest biomes. The work is an excellent contribution to the field of macro ecology and will help inform aspects of plant scaling relationships and sensitivities to climate.

Response: Thank you very much for your kind words and for reviewing our paper.

Comment: In general, the authors are careful to not draw definitive conclusions about the generalized trends they found in the data, but rather offer substantial evidence for each claim to support the major observations regarding plant trait sensitivity. The one area of the work that probably needs a bit more discussion is the role of uncertainty and variation in the findings. Some of the analyses do not explicitly incorporate uncertainty or variation in the estimates. Since the approaches here rely on averaged remotely sensed climate estimates, we can expect there is a bit more uncertainty in the final estimates than reported. That said, I think the authors are taking a reasonable approach and adding propagated and quantified uncertainty estimates are unlikely to change the final findings.

Response: The issue of variability and plasticity within species is an important one that we should have addressed more directly in the previous version of the manuscript. We agree that this is important both from a methodological standpoint (uncertainty propagation) and an ecological one (i.e., what drives variability and plasticity in crown architectural traits within species?). First, it's important to clarify why we decided to focus our analysis on differences in crown architecture among species. Our motivation stems from previous work where we showed that variability in crown architecture among trees is predominantly driven by differences between species rather than plasticity within them (see Case Study 2 in Jucker et al. 2022 GCB). This is why we chose to focus explicitly on characterising the crown architecture of a large number of tree species (nearly 2000) distributed across all biomes, as previous work on this topic has focused either on comparing a small number of temperate species (10-20) across multiple sites (e.g., Lines et al. 2012 GEB) or a slightly larger number of species (100-200) sampled at single site in the tropics (e.g., Iida et al. 2011 FEcol). We have made sure to better articulate this motivation in the revised manuscript (L646-650).

The reason why we then chose to limit our analysis to exclusively look at differences between species (ignoring variability within them) has to do with the nature of our data. Specifically, many species in our analysis have insufficient data to appropriately quantify intraspecific variability in both their architecture and climatic envelopes: 314 species (~16%) were measured at a single site and only 342 species (<20%) were sampled widely enough to characterise their architectural plasticity along environmental gradients (the latter are the subset we analysed for Case Study 2 in Jucker et al. 2022 GCB). Any estimates of variability and uncertainty we provide would be strongly influenced by differences in sample size among

species (which as we highlight in the paper vary from 10 to 22,835 individuals per species). In revising the paper, we have made sure to more clearly highlight this limitation (L650-653).

In terms of our methodology, one thing we did do to explicitly mitigate uncertainty linked to within-species variability is to only retain samples from a species' dominant biome (i.e., in cases where a species was recorded in multiple biomes, we only kept data from the biome where it was found most frequently; see L594-597 for details). As you suggested, for each species we have now also calculated the standard deviations of the various environmental predictors used in our analysis. These are reported in the updated species-level data frame (species_data.csv) that we have provided as part of the data and code archive accompanying our paper. However, we caution on overinterpreting these standard deviation values, as for many species they will considerably underestimate the true range of environmental conditions within which they can grow.

Finally, to better integrate the question of within-species variability in our paper, we have added a new section to the discussion (L536-554) where we highlight this as an important area for future research. Here, in addition to highlighting some of the limitations of our data to address the questions of intraspecific variability, we have also made an effort to highlight several concrete future research questions which would complement and build on our findings.

Comment: I see no issues regarding data analysis, interpretation or conclusions. In general, the authors had a robust analysis approach that provided insight on the specific hypotheses tested. As stated above, a structured approach to uncertainty propagation would be appreciated, but would be unlikely to change the overall findings of the study.

Response: Thank you, we are pleased that overall the methodology and the interpretations we draw from our results come across as robust. See our previous response for how we have addressed the issue of intraspecific variability in our analysis.

Comment: The methodology is sound and the approaches are in line with expectations for this type of study. My one suggestion would be for the authors to incorporate uncertainty in the analyses more explicitly. For instance, (if I'm not mistaken) the mean of specific climate/ecosystem variables was used for entire species, when these species may exist across a range of growing conditions. In other words, variance exists that may be important in species that are generalists and exist in many places.

Response: We agree and as mentioned above we now provide estimates of environmental variability for each species as part of the data and code archive accompanying our paper.

Comment: Enough detail is provided for the methods to be reproduced and code for the analysis is provided. I hope the authors will publish the final cleaned and assembled dataset for others to use.

Response: Thank you. We have revised the data and code archive accompanying our paper to include all data necessary to calculate the species-level estimates reported in our main analysis. We note however, that these are just a subset of the larger Tallo database, which includes additional tree-level records not included in our present analysis (e.g., trees from species with fewer than 10 individuals within a given biome, or trees where species IDs were missing). The Tallo database is permanently archived on Zenodo (<https://zenodo.org/records/6637599>).

Comment: Line 223: A nice, simple introduction that is concrete.

Response: Thank you!

Comment: Line 231: Succinct explanation that covers the main applications to remote sensing and vegetation models. Great.

Response: Thank you, we are pleased these applications came across clearly.

Comment: Line 243: Solid summary of the main research on this topic.

Response: Thank you!

Comment: Line 266: Excellent sample size. The presented figure of the distribution of sample locations should be colored to show number of samples per aggregate unit. This would be a more transparent way of showing the distribution of individual trees in the dataset. It would also be useful to show the aggregate number of samples as a distribution over the latitude and longitude of the global map. The second suggestion is less essential, but would provide some more understanding of the data distribution than the presence / absence map as it is shown now.

Response: Thank you and great suggestion. We have revised the map in Fig. 1 so that the colour of the grid cells reflects the number of samples acquired from each location.

Comment: Lines: 275-284 Will this compiled and standardized database be made available somewhere? The location should be cited.

Response: Yes, this will be archived on Zenodo and we have provided a link to the DOI in the revised manuscript (note the archive will be made public following the review of this paper). As noted above, these are just a subset of the larger Tallo database, which includes additional tree-level records not included in our present analysis. The Tallo database is already permanently archived on Zenodo (<https://zenodo.org/records/6637599>).

Comment: Lines 299-311I appreciate taking different analysis approaches, but this first one seems a bit arbitrary and it's unclear if it necessarily adds anything useful (compared to the second approach) to the analysis. Maybe introducing this section with a justification for why this simpler approach is used.

Response: We agree that the first approach is more arbitrary than the second, but felt it was valuable to retain both methods as they are complementary. In particular, the first method where height, crown diameter and crown aspect ratio are estimated for a tree of fixed size ($D = 30$ cm in our case) has been used extensively in previous studies (e.g., Iida et al. 2011 FEcol; Lines et al. 2012 GEB; Poorter et al. 2012 PlantEcol; Jucker et al. 2022 GCB) – meaning readers will be familiar with this approach. It also has the advantage that the predicted values of height, crown diameter and crown aspect ratio are in units that are easily interpretable (e.g., meters), making it much simpler for readers to get a sense of the magnitude of the effects we report. Ultimately as we show in Supplementary Fig. 1 the two methods give nearly identical results, so we preferred to report both for the benefit of the reader.

Comment: Lines: 360 Please provide detail on why 2008 was chosen as the baseline year.

Response: We chose this year as it corresponds with the period in which most of the allometric data were collected (although we do note that exact dates are not known for all our records). We have clarified this in the text.

Comment: Line 373: For each of these species have you reported the mean and standard deviation of your estimates (e.g. mean and standard deviation of precipitation for a single species, etc.)? If not please report them in a table in the supplement. This will provide an estimate of the uncertainty / variation across species (especially those species that exist in many different growing environments).

Response: Agreed. As mentioned above these are now reported in the updated species-level data frame (species_data.csv) that we have provided as part of the data and code archive accompanying our paper.

Comment: Line 1104: It would be helpful if the colors were ordered as shown in the legend on the top of the figure. For instance, group all purple, grey, and yellow/red into a block. The colors and orders of colors are seemingly grouped by the "low, med, high" groupings, but I'm not sure if this is easiest for interpretation of the proportions of each class in each biome.

Response: This is a great suggestion, we have updated the figure accordingly!

Comment: Looking forward to seeing the published version of this work! Atticus Stovall

Response: Thank you!

Comment: Reviewer #1 remarks on code availability: I was able to download and run the majority of the code provided. No README was included but the comments were helpful for understanding the different analysis sections. I was unable to evaluate any of the tree-level analysis because these data were not provided, but I imagine the dataset is quite large.

Response: We have now provided a README file to accompany the data and core archive, as well as tree-level data and code used to estimate the species-level parameters reported in the main text.

Reviewer #2

Comment: This a timely article and it seeks to answer urgent questions on tree allometry. However, authors need to clarify the data analysis.

Response: Thank you for reviewing our paper, we are pleased you enjoyed it.

Comment: What are the noteworthy results? Tree heights were constrained by light competition and water availability while exposure to wind and fire constrained crown size.

Response: Thank you, we are pleased the key results were easy to distil.

Comment: Will the work be of significance to the field and related fields? How does it compare to the established literature? If the work is not original, please provide relevant references. Yes. It is unclear whether reported scaling exponents conformed to theoretical predictions. Given the large dataset, it will be interesting to investigate if scaling exponents of the various allometries tested are in line with existing theoretical predictions such as the stress similarity model, metabolic scaling theory, geometric similarity model.

Response: This is a good observation, but the reason why we did not present this analysis here is that it has been carried out previously using the entire Tallo database (see Case Study 1 in Jucker et al. 2022 GCB).

Comment: Does the work support the conclusions and claims, or is additional evidence needed?

Yes once the corrections are made

Response: Thank you.

Comment: Are there any flaws in the data analysis, interpretation and conclusions? Do these prohibit publication or require revision? Maybe. Allometry parameters can widely vary with regression techniques. What was the rationale for using log-transformed data? Secondly, selection of allometric models follows a common scientific procedure where different models

are experimented and tested, and various indices/coefficient (such R, adjustedR2, RMSE, AIC or BIC, etc.) used in selecting the model that best fits or explains variances in the data. I am seriously questioning how the authors arrived at opting for the log-log regression model. If this model performance was better than other models considered in your analysis? What other models were considered and assessed? These details are necessary to understand the analysis and information reported. I am questioning whether there might be other models which could have fitted the data better than the log-log regression model?

Response: This is a very good point and something we should have dealt with in more detail in the previous version of our paper. Conceptually, the advantage of the log-log model (i.e., a power-law) lies in its simplicity and the fact that it has a long history of being used to study allometry, thus directly linking our work to a large body of existing literature). However, we fully agree that testing the appropriateness of the log-log is something we should have done more formally before adopting it for our analysis. This is especially true for modelling height–diameter scaling relationships, where previous work has suggested that other functional forms (particularly saturating ones like the Michaelis-Menten functions) may provide a better fit to the data. To test this we used a subset 754 of well sampled species in our datasets (minimum sample size = 50 trees per species; minimum stem diameter range = 30 cm). For each species we took a random subset of 50 individuals to ensure species with larger sample sizes would not dominate the signal. We then used the same approach described in the main text to estimate HRESID and HD=30 for each species using both a power-law and a Michaelis-Menten function. We found that both estimates of HRESID ($\rho = 0.97$) HD=30 ($\rho = 0.93$) derived from a power-law and a Michaelis-Menten function were strongly correlated (see Supplementary Fig. 2). This indicates that the choice of functional form used to model the scaling relationship had little to no effect on our results. Moreover, of the two functional forms the power-law model fit the data considerably better (based on both RMSE and AIC). This was true both when modelling the entire dataset together to estimate HRESID (RMSE = 5.4 m and 5.8 m for power-law and Michaelis-Menten, respectively) and when accounting for variation in allometric scaling relationships among species to estimate HD=30 (RMSE = 3.5 m and 3.9 m for power-law and Michaelis-Menten, respectively). Based on this we opted to model height–diameter relationships using a power-law function. This model comparison is provided in Supplementary Fig. 2 and discussed in the Methods section of the revised paper. We have also provided R code to replicate this comparison as part of the data and code archiving package uploaded alongside our paper.

Comment: What are the merits of the transformed log-log model? Why did it perform better in explaining variance in the data used for your analyses? I am curious if the log-log model best explained the allometry of all the tree species in this study? Can you provide supporting information or discussions to clarify this?

Response: See our response above about this point. As mentioned, we have now compared the log-log height–diameter model to an alternative saturating function, and found that the log-log model actually fit the data best. Note that we did not perform this comparison for the scaling relationships between crown diameter–diameter and crown diameter–height, as here the literature is much more in agreement on the appropriateness of a log-log model to capture these relationships (e.g., see Loubota Panzou et al. 2020 GEB; Shenkin et al. 2020 Front. For. Glob. Change; Jucker et al. 2022 GCB), and no obvious alternative candidate models have been proposed.

Comment: In general, SMA estimates of lines summarizing relationship between two variables are superior to ordinary least square linear regression and have been advocated in the literature because residual variance is minimized in both the X and Y dimensions rather

than in the Y dimension only. When comparing regressions, differences can occur in either exponent of a (Y intercept) and/or b (regression slope). If b differs among species, species with larger b will have greater increase in Y per increment of X. If a differs, but b does not, species with larger a will have a consistently larger amount of Y at any given value of X (Kohyama and Hotta, 1990; Sposito and Santos, 2001). Authors should clarify how the ordinary least square regression was suitable for the analysis.

Response: This is another aspect we should have clarified better in methods. As the reviewer correctly points out, there is a lot of debate in the allometry literature about the appropriateness of ordinary least squares (OLS) regression as opposed to model II type approaches such as standardised major axis (SMA) regression for modelling allometric scaling relationships. It is therefore important to justify why we chose one approach over the other. In our case this comes down to the objective of the analysis and whether the models are being used to estimate scaling coefficients or generate predictions. When estimating scaling coefficients (i.e., the slope of the log-log regression), then SMA regression is typically the preferred choice. However, the goal of our analysis was not to compare species based on their scaling coefficients, but instead to use the models to generate predictions of height, crown area and crown aspect ratio that we could then use to compare species. In this case, the advice is to always use OLS regression. This point is nicely summarised by Pierre Legendre, who developed the user guide and statistical package for model II regression in R: “When the purpose of the study is not to estimate the parameters of a functional relationship, but simply to forecast or predict values of y for given x’s, use OLS in all cases” (see recommendation 6 here: <https://cran.r-project.org/web/packages/lmodel2/vignettes/mod2user.pdf>). We have now explicitly justified our modelling choice in the methods of the revised paper (L614-616).

Comment: Is the methodology sound? Does the work meet the expected standards in your field?

Yes

Response: Thank you.

Comment: Is there enough detail provided in the methods for the work to be reproduced?

Yes

Response: Thank you again for reviewing our paper and for the helpful feedback!

Reviewer #3

Comment: Jucker et al. provide and analyze a prodigious dataset on the global relationship of tree height (H), crown diameter (D), and crown aspect ratio (D/H) to climate, disturbance, competition, other functional traits, phylogeny and each other. Height and height scaling relationships appear to be most strongly affected by water availability and competition; crown width, by exposure to disturbance by wind or fire. Some lineages had unusual crown forms apparently tied more to phylogeny than environment.

Response: Thank you for reviewing our paper.

Comment: This is an important paper that quantifies and crystallizes several patterns evident, in many cases, to natural historians for decades if not centuries. The global nature of the database, the extraordinary range of plots, species, and evolutionary lineages assessed, the sound statistical frameworks (in most cases) employed to analyze the data, and the conclusions reached will likely make this paper a touchstone for years to come. The authors are to be congratulated.

Response: Thank you for the kind words, we are very pleased you enjoyed the paper.

Comment: However, as currently written, the manuscript has two central, interrelated flaws: a lack of a conceptual framework to predict or account for patterns in tree height, diameter, and crown aspect ratio, and a failure to cite some centrally important papers that provide such a framework. These flaws must be addressed.

Response: We fully agree that the relationships and predictions we test in our paper need to be well anchored to the existing literature if our study is to provide a comprehensive overview of the drivers that shape the crown architecture of trees. We really appreciate the suggestions for improving this aspect of our work and have incorporated them into our revision (see responses below for details).

Comment: Specifically, the authors should first cite Givnish et al. 2015 (*Ecology* 95: 2991-3007, Determinants of maximum tree height in Eucalyptus species along a rainfall gradient in Victoria, Australia) AND briefly summarize their general model relating maximum tree height to climate, soil fertility, and disturbance, to the allometry of allocation to wood vs. leaves with height, and to vertical stratification of environmental conditions with height above the ground. That paper shows a strong relationship of tree height to climatic moisture supply relative to demand, as quantified by P/E_p (mean annual precipitation divided by pan evaporation) – which is closely related to the aridity index used by Jucker et al. – and argues for its generality, albeit modulated by differences in temperature, latitude, soil fertility, and disturbance. These theoretical and empirical findings are directly relevant to the subject of the current paper, and they really must be cited – not as a content-less superscript, but for specific predictions and statistical relationships obtained along a substantial climatic gradient. Those earlier findings are directly relevant to what is arguably the most important conclusions of the current paper. Given that the authors tie global variation in height and crown aspect ratio (tied to height) to leaf nitrogen content and specific leaf area, they should also cite Givnish et al. 2015 for both their model predictions and findings of such relationships in their Eucalyptus study ... and for the climatic and edaphic drivers of variation in leaf N and SLA. They might also consider citing Smith et al. 2023 (*Nature Communications* 14: 7173, Ecophysiological adaptations shape distributions of closely related trees along a climatic moisture gradient) that detail shifts in 50 functional traits, as well as height growth, mass growth, and survival in 10 species of Eucalyptus grown in common gardens along the same gradient. That paper shows that species differences in many of those traits are heritable and largely accord with economic models to maximize rates of height growth.

Response: Thank you for this, it is really helpful both in terms of framing the analysis of variation in tree height along aridity gradients, but also relating this back to other functional traits such as leaf nitrogen content. We have now incorporated Givnish et al. 2015 into the text, including highlighting it in the introduction and citing it as a key reference supporting our predictions outlined in Table 1. We have also refer back to it at several points in the discussion, citing it as a key example supporting our findings about how investment in height growth varies along aridity gradients and how we might expect this to covary with investment in leaf nitrogen.

Comment: Second, the authors should cite King 1981 (*Oecologia* 51: 351-356, Tree dimensions: Maximizing the rate of height growth in dense stands) and Givnish 1995 (*Plant stems: biomechanical adaptations for energy capture and influence on species distributions*. Pp. 3-49 in B. L. Gartner (ed.), *Plant Stems: Physiology and Functional Morphology*. Chapman and Hall, New York). These papers explain why the ratio of tree crown width to height should converge on a specific value in crowded stands. The first paper presents a quantitative model for the optimal allocation of energy between crown and bole; the second shows that the optimal value of that ratio corresponds to crown width increasing in proportion to crown height (i.e., crown width/height ratio is constant) and accounts for the observed relationship in fully

stocked (crowded) stands documented by O'Neill & DeAngelis 1981 (Pp. 411-449 in DE Reichle (ed), *Dynamic Properties of Forest Ecosystems*, Cambridge University Press) and used by Givnish 1986 (*Journal of Theoretical Biology* 119: 139-146, Biomechanical constraints on self-thinning in plant populations) to provide the first mechanistic explanation for the $-3/2$ power thinning law in plants. Givnish 1995 provides an extensive table of predictions for how difference aspects of crown form should vary with environmental conditions. Arguments for broader crowns relative to height in shade-adapted trees are presented, as well as supporting data from an extensive dataset on North American champion trees.

Response: Again, thank you for highlighting this important work. We agree that it's really important to link our finding back to earlier work that set the foundations for our current understanding the biophysical constraints that shape tree architecture. The papers by King 1981 and Givnish 1995 are particularly relevant to the opening two paragraph of the introduction, where we provide a broad overview of the various constraints that trees operate under when expanding their crowns both vertically and horizontally. We have also incorporated them in Table 1, where they help support our hypotheses about how we might expect tree height and crown width to vary in relation to competition for light with neighbours. Finally, they help support several of our key findings, and we have therefore refereed to them specifically in the revised discussion.

Comment: Third, while the authors' statistical approaches are generally sound, I note two weaknesses that they should either justify or redress: (A) In studies involving allometric variation (e.g., between crown height and width), Type I regression models that minimize the residual variance of the dependent variable based on relationships to independent variables – i.e., ordinary linear regressions, as employed here) are often viewed as inappropriate, given that there is no clearly distinction between dependent and independent variables. Type II regressions, which minimize the residual variance about the regression in both the x- and y-directions, have generally been employed in such cases. The authors should justify their use of Type I regression or replace it with Type II regression. I am not sure whether a Type II version of generalized linear models is available however, and so they may have to use the current version of GLM for those (important) analyses. (B) It's generally recognized that phylogenetic regression – which accounts for relationships among species, and thus the non-independence of data points – should be used in comparative studies like those at the heart of this paper. I recommend that the authors should at least present phylogenetic regressions that complement their traditional regressions, even if they rely mainly on the latter. They already have relationships in hand for many of their study species, as shown by Figure 3.

Response: Thank you for highlight both of these points. Relating to point (A) about the appropriateness of Type I vs Type II regression, we agree that we should have better justified our choice for the former. As the reviewer correctly points out, there is a lot of debate in the allometry literature about the appropriateness of Type I regression (e.g., ordinary least squares, OLS) as opposed to Type II approaches such as standardised major axis (SMA) regression for modelling allometric scaling relationships. It is therefore important for us to justify why we chose one approach over the other. In our case this comes down to the objective of the analysis – specifically whether the models are being used to estimate scaling coefficients or generate predictions. When estimating scaling coefficients (i.e., the slope of the log-log regression), then SMA regression is typically the preferred choice. However, the goal of our analysis was not to compare species based on their scaling coefficients, but instead to use the models to generate predictions of height, crown area and crown aspect ratio that we could then use to compare species. In this case, the advice is to always use OLS regression, especially as the structure of our models requires incorporating random effects which are not supported by SMA. This point is nicely summarised by Pierre Legendre, who developed the

user guide and statistical package for model II regression in R: “When the purpose of the study is not to estimate the parameters of a functional relationship, but simply to forecast or predict values of y for given x 's, use OLS in all cases” (see recommendation 6 here: <https://cran.r-project.org/web/packages/lmodel2/vignettes/mod2user.pdf>). We have now explicitly justified our modelling choice in the methods of the revised paper (L614-616).

Regarding point (B) about accounting for phylogeny in our analysis, we fully agree and this is actually something we have already done by using phylogenetic generalised least squares (PGLS) regressions for the main analyses presented in our study (i.e., the results presented in Figs 4 & 5 of our paper). We have made sure to make this point clearer in revising our paper (see L779-786). We also note that the underlying models used to estimate species-level crown attributes also leverage phylogenetic relationships among species by incorporating species, genus and family as nested random effects. This approach is very similar (and in some cases preferable) to using PGLS (see here for an in-depth discussion: <https://statmodeling.stat.columbia.edu/2016/02/14/hierarchical-models-for-phylogeny-heres-what-everyones-talking-about/>).

Comment: Fourth, I strongly recommend that the authors revise Figure 3 so as to increase its impact, by (A) stretching it vertically a bit, to give better resolution of individual branches, and then (B) using maximum likelihood mapping to colorize branches to visualize the evolution of tree height, crown width, and crown aspect ratio. Widely available R packages by Liam Revell can be used for this task.

Thomas J. Givnish

Henry Allan Gleason Professor of Botany

Wilhelm Hofmeister Professor of Botany

University of Wisconsin-Madison

Response: We agree that in Fig. 3 the separation between individual branches is hard to see. But unfortunately this is product of the large number of species represented on the phylogenies. We attempted to stretch the phylogenies vertically as suggested, but this did not help resolve this issue, as there are simply too many species. As for suggestion (B), we did also try this in a previous iteration of the figure, but again found that because of the large number of species it ended up being more confusing than helpful. We were also unsure about representing these ancestral reconstructions on the phylogenies, as they are not the focus of our analysis and we felt they may confuse the reader. We therefore opted to keep Fig. 3 as is, but thank the reviewer for their helpful suggestions on how to improve it further.

Reviewer #1

Comment: The revisions in the current version of the paper addressed all of my previous comments. I am satisfied with the current version and believe it is ready for publication. Nice work.

Response: Thank you very much for reviewing our paper, we are glad you enjoyed it.

Reviewer #2

Comment: The authors did a fantastic job clarifying their analysis and methods and providing supplementary files. I recommend it should be accepted.

Response: Thank you for very much for the kind words and for reviewing our paper.

Reviewer #3

Comment: Jucker et al. have responded adequately to most of the reviewers' comments. Their efforts are, in general, to be applauded.

Response: Thank you for reviewing our paper a second time and for the positive feedback.

HOWEVER, they have failed to respond to my central criticism – that they need to provide a simple conceptual framework for ecological trends in tree height and crown width vs. height. Worse, their rebuttal letter misrepresents what they actually did in editing the manuscript. The manuscript text must be changed. As detailed below, the changes needed are minimal – rewriting three sentence – but crucial. To quote their letter on two key points:

1. COMMENT: Specifically, the authors should first cite Givnish et al. 2015 (Ecology 95: 2991-3007, Determinants of maximum tree height in Eucalyptus species along a rainfall gradient in Victoria, Australia) AND briefly summarize their general model relating maximum tree height to climate, soil fertility, and disturbance, to the allometry of allocation to wood vs. leaves with height, and to vertical stratification of environmental conditions with height above the ground. That paper shows a strong relationship of tree height to climatic moisture supply relative to demand, as quantified by P/E_p (mean annual precipitation divided by pan evaporation) – which is closely related to the aridity index used by Jucker et al. – and argues for its generality, albeit modulated by differences in temperature, latitude, soil fertility, and disturbance. These theoretical and empirical findings are directly relevant to the subject of the current paper, and they really must be cited – not as a content-less superscript, but for specific predictions and statistical relationships obtained along a substantial climatic gradient. Those earlier findings are directly relevant to what is arguably the most important conclusions of the current paper. Given that the authors tie global variation in height and crown aspect ratio (tied to height) to leaf nitrogen content and specific leaf area, they should also cite Givnish et al. 2015 for both their model predictions and findings of such relationships in their Eucalyptus study ... and for the climatic and edaphic drivers of variation in leaf N and SLA. They might also consider citing Smith et al. 2023 (Nature Communications 14: 7173, Ecophysiological adaptations shape distributions of closely related trees along a climatic moisture gradient) that detail shifts in 50 functional traits, as well as height growth, mass growth, and survival in 10 species of Eucalyptus grown in common gardens along the same gradient. That paper shows that species differences in many of those traits are heritable and largely accord with economic models to maximize rates of height growth.

RESPONSE: Thank you for this, it is really helpful both in terms of framing the analysis of variation in tree height along aridity gradients, but also relating this back to other functional traits such as leaf nitrogen content. We have now incorporated Givnish et al. 2015 into the text, including highlighting it in the introduction and citing it as a key reference supporting our predictions outlined in Table 1. We have also refer back to it at several points in the discussion, citing it as a key example supporting our findings about how investment in height growth varies along aridity gradients and how we might expect this to covary with investment in leaf nitrogen."

Not only do the authors not "highlight" the conceptual framework of Givnish et al. 2015 in the introduction, they do not cite it all, let alone give a simple sentence summarizing its key conceptual framework re what determines tree height. Furthermore, I fundamentally disagree with their explanation for why height growth along aridity gradients should covary with leaf nitrogen. This is unacceptable, both for materially ignoring the integrative conceptual framework of Givnish et al., and for stating things in their rebuttal letter that are simply untrue. The authors have stuck to their original approach and not changed their intro in any way (based on the tracked-changes version), except to add a final paragraph about some trends in crown form they document later.

Response: Just to briefly clarify, we did in fact cite Givnish et al. 2015 in the introduction. It is one of the key papers we highlight in Table 1, which though currently located at the end of the manuscript will appear in print in the introduction where it first mentioned. We also cite this paper in the discussion, where we mention its findings explicitly in the context of Australian forests. To avoid any confusion we have now also cited the paper in the main text of the introduction as well (ref 44 in the revised manuscript).

I recommend the authors remedy this situation – and their failure to response to my second main criticism (see point 2 below) by inserting the following two sentences on line 239:

Maximum tree height should increase toward moister or more fertile sites, based on the balance between photosynthesis vs. increases with height in allocation to unproductive stem tissue and hydraulic limitations to photosynthesis. The ratio of crown diameter to height that maximizes upward growth should not vary with tree height, but relative crown width should be greater in more open habitats (or under closed canopies) where height growth increases energy capture less than horizontal crown expansion.

Reference A: Givnish, T. J., Wong, S. C., Stuart-Williams, H., Holloway-Phillips, M. & Farquhar, G. D. Determinants of maximum tree height in Eucalyptus species along a rainfall gradient in Victoria, Australia. *Ecology* 95, 2991–3007 (2014).

Rationale: (1) The authors need to cite and highlight the Givnish et al. 2015 framework, based on my earlier criticism and their own statements in the rebuttal letter. (2) Earlier in the intro paragraph, they make the conceptually erroneous statement that "For instance, in arid climates woody biomass allocation tends to shift away from height growth to limit the risk of hydraulic failure, resulting in trees that are shorter for a given diameter." As Givnish et al. 2015 showed, including only hydraulic limitation does not account for shifts in tree height with increasing aridity; the allometric variation in allocation to unproductive stem tissue must also be included. (3) The second recommended sentence addresses my second main criticism (see next paragraph) and provides a pointer to the conceptual framework for relative crown width.

Response: Thank you for clarifying these points. We have adapted the suggested text and incorporated it into both the revised introduction (L232–238) and discussion (L453–455). In

both cases we have taken care to explicitly cite Givnish et al. (2014). In particular, this section of the introduction now reads:

For instance, in arid climates woody biomass allocation tends to shift away from height growth and towards crown expansion to limit the risk of hydraulic failure and maximise energy capture, resulting in trees that are shorter for a given diameter and have wider crown profiles. Conversely, when water and nutrients are non-limiting to photosynthesis, strong competition for light leads to greater investment in height growth and relative allocation of carbon to woody tissues, pushing trees closer to their structural and hydraulic safety margins.

2. COMMENT: Second, the authors should cite King 1981 (*Oecologia* 51: 351-356, *Tree dimensions: Maximizing the rate of height growth in dense stands*) and Givnish 1995 (*Plant stems: biomechanical adaptations for energy capture and influence on species distributions*. Pp. 3-49 in B. L. Gartner (ed.), *Plant Stems: Physiology and Functional Morphology*. Chapman and Hall, New York). These papers explain why the ratio of tree crown width to height should converge on a specific value in crowded stands. The first paper presents a quantitative model for the optimal allocation of energy between crown and bole; the second shows that the optimal value of that ratio corresponds to crown width increasing in proportion to crown height (i.e., crown width/height ratio is constant) and accounts for the observed relationship in fully stocked (crowded) stands documented by O'Neill & DeAngelis 1981 (Pp. 411-449 in DE Reichle (ed), *Dynamic Properties of Forest Ecosystems*, Cambridge University Press) and used by Givnish 1986 (*Journal of Theoretical Biology* 119: 139-146, *Biomechanical constraints on self-thinning in plant populations*) to provide the first mechanistic explanation for the $-3/2$ power thinning law in plants. Givnish 1995 provides an extensive table of predictions for how difference aspects of crown form should vary with environmental conditions. Arguments for broader crowns relative to height in shade-adapted trees are presented, as well as supporting data from an extensive dataset on North American champion trees.

RESPONSE: Again, thank you for highlighting this important work. We agree that it's really important to link our finding back to earlier work that set the foundations for our current understanding the biophysical constraints that shape tree architecture. The papers by King 1981 and Givnish 1995 are particularly relevant to the opening two paragraph of the introduction, where we provide a broad overview of the various constraints that trees operate under when expanding their crowns both vertically and horizontally. We have also incorporated them in Table 1, where they help support our hypotheses about how we might expect tree height and crown width to vary in relation to competition for light with neighbours. Finally, they help support several of our key findings, and we have therefore refereed to them specifically in the revised discussion.

Again, the authors did not – so far as I can see – insert any statement addressing the importance of these papers and the ideas and data they present for setting relative crown width. The second sentence I proposed adding (see point 1) would remedy that.

Response: Note again that both of these papers were in fact cited throughout our revised manuscript, including the main text of the introduction, Table 1 and the discussion (they are Refs 32 and 7, respectively). As mentioned above we have now amended the introduction to explicitly mention the prediction that trees should invest more (in relative terms) in crown expansion vs height growth in dry, open environments (as suggested in Point 1).

3. Finally, I fundamentally disagree with the authors' explanation for why tree height should increase with leaf nitrogen content (lines 505-508). They ignore the effect of increased

photosynthesis per unit leaf mass (which increases with leaf N content) on maximum tree height. They ignore the effect of lowered allocation to roots in causing greater tree height on moister or more fertile sites AND on increasing whole-plant shade tolerance (see reference 7). The latter undercuts their argument for increased allocation to height growth ... and, of course, fails to account for decreased whole-plant shade tolerance with height (a key point made by reference 7). The authors have no data showing an increased investment in height growth at a given height on sites with higher leaf N content. Citing our paper to implicitly endorse their argument makes me uncomfortable. To cope with these problems, I recommend simply-writing the problematic lines 505-508 as follows:

Based on this we would expect species with higher leaf nitrogen to invest more in both height growth and crown expansion to allow them to intercept more light, limit self-shading and optimise the distribution of leaves across their crowns. In addition, higher leaf nitrogen content should lead to greater photosynthetic rates and result in greater maximum tree height – as found previously along a gradient of increasing rainfall in Australia.

As again is Givnish et al. 2015 (see above).

If the authors make the two changes listed above, under points 1 and 3, I would fully support publication of their important manuscript.

Response: Thank you for clarifying this point, we have revised the text as suggested. This now reads as follows:

Moreover, higher leaf nitrogen content should lead to greater photosynthetic rates and investment in height growth – as found previously along a rainfall gradient in Australia

Jucker et al. have responded adequately to most of the reviewers' comments. Their efforts are, in general, to be applauded.

HOWEVER, they have failed to respond to my central criticism – that they need to provide a simple conceptual framework for ecological trends in tree height and crown width vs. height. Worse, their rebuttal letter misrepresents what they actually did in editing the manuscript. The manuscript text **must** be changed. As detailed below, the changes needed are minimal – rewriting three sentence – but crucial.

To quote their letter on two key points:

1. “**Comment:** Specifically, the authors should first cite Givnish et al. 2015 (Ecology 95: 2991-3007, Determinants of maximum tree height in Eucalyptus species along a rainfall gradient in Victoria, Australia) AND briefly summarize their general model relating maximum tree height to climate, soil fertility, and disturbance, to the allometry of allocation to wood vs. leaves with height, and to vertical stratification of environmental conditions with height above the ground. That paper shows a strong relationship of tree height to climatic moisture supply relative to demand, as quantified by P/E_p (mean annual precipitation divided by pan evaporation) – which is closely related to the aridity index used by Jucker et al. – and argues for its generality, albeit modulated by differences in temperature, latitude, soil fertility, and disturbance. These theoretical and empirical findings are directly relevant to the subject of the current paper, and they really must be cited – not as a content-less superscript, but for specific predictions and statistical relationships obtained along a substantial climatic gradient. Those earlier findings are directly relevant to what is arguably the most important conclusions of the current paper. Given that the authors tie global variation in height and crown aspect ratio (tied to height) to leaf nitrogen content and specific leaf area, they should also cite Givnish et al. 2015 for both their model predictions and findings of such relationships in their Eucalyptus study ... and for the climatic and edaphic drivers of variation in leaf N and SLA. They might also consider citing Smith et al. 2023 (Nature Communications 14: 7173, Ecophysiological adaptations shape distributions of closely related trees along a climatic moisture gradient) that detail shifts in 50 functional traits, as well as height growth, mass growth, and survival in 10 species of Eucalyptus grown in common gardens along the same gradient. That paper shows that species differences in many of those traits are heritable and largely accord with economic models to maximize rates of height growth.

Response: Thank you for this, it is really helpful both in terms of framing the analysis of variation in tree height along aridity gradients, but also relating this back to other functional traits such as leaf nitrogen content. We have now incorporated Givnish et al. 2015 into the text, including highlighting it the introduction and citing it as a key reference supporting our predictions outlines in Table 1. We have also refer back to it at several points in the discussion, citing it as a key example supporting our findings about how investment in height growth varies along aridity gradients and how we might expect this to covary with investment in leaf nitrogen.”

My response: Not only do the authors not “highlight” the conceptual framework of Givnish et al. 2015 in the introduction, they do not cite it all, let alone give a simple sentence summarizing its key conceptual framework re what determines tree height. Furthermore, I fundamentally disagree with their explanation for why height growth along aridity gradients should covary with leaf nitrogen.

This is unacceptable, both for materially ignoring the integrative conceptual framework of Givnish et al., and for stating things in their rebuttal letter that are simply untrue. The authors have stuck to their original approach and not changed their intro in any way (based on the tracked-changes version), except to add a final paragraph about some trends in crown form they document later.

I recommend the authors remedy this situation – and their failure to respond to my second main criticism (see point 2 below) by inserting the following two sentences on line 239:

Maximum tree height should increase toward moister or more fertile sites, based on the balance between photosynthesis vs. increases with height in allocation to unproductive stem tissue and hydraulic limitations to photosynthesis^A. The ratio of crown diameter to height that maximizes upward growth should not vary with tree height, but relative crown width should be greater in more open habitats (or under closed canopies) where height growth increases energy capture less than horizontal crown expansion^{7,32}.

Reference A: Givnish, T. J., Wong, S. C., Stuart-Williams, H., Holloway-Phillips, M. & Farquhar, G. D. Determinants of maximum tree height in *Eucalyptus* species along a rainfall gradient in Victoria, Australia. *Ecology* **95**, 2991–3007 (2014).

Rationales: (1) The authors need to cite and highlight the Givnish et al. 2015 framework, based on my earlier criticism and their own statements in the rebuttal letter. (2) Earlier in the intro paragraph, they make the conceptually erroneous statement that “For instance, in arid climates woody biomass allocation tends to shift away from height growth to limit the risk of hydraulic failure, resulting in trees that are shorter for a given diameter.” As Givnish et al. 2015 showed, including only hydraulic limitation does not account for shifts in tree height with increasing aridity; the allometric variation in allocation to unproductive stem tissue must also be included. (3) The second recommended sentence addresses my second main criticism (see next paragraph) and provides a pointer to the conceptual framework for relative crown width.

2. Comment: Second, the authors should cite King 1981 (*Oecologia* 51: 351-356, Tree dimensions: Maximizing the rate of height growth in dense stands) and Givnish 1995 (Plant stems: biomechanical adaptations for energy capture and influence on species distributions. Pp. 3-49 in B. L. Gartner (ed.), *Plant Stems: Physiology and Functional Morphology*. Chapman and Hall, New York). These papers explain why the ratio of tree crown width to height should converge on a specific value in crowded stands. The first paper presents a quantitative model for the optimal allocation of energy between crown

and bole; the second shows that the optimal value of that ratio corresponds to crown width increasing in proportion to crown height (i.e., crown width/height ratio is constant) and accounts for the observed relationship in fully stocked (crowded) stands documented by O'Neill & DeAngelis 1981 (Pp. 411-449 in DE Reichle (ed), Dynamic Properties of Forest Ecosystems, Cambridge University Press) and used by Givnish 1986 (Journal of Theoretical Biology 119: 139-146, Biomechanical constraints on self-thinning in plant populations) to provide the first mechanistic explanation for the $-3/2$ power thinning law in plants. Givnish 1995 provides an extensive table of predictions for how different aspects of crown form should vary with environmental conditions. Arguments for broader crowns relative to height in shade-adapted trees are presented, as well as supporting data from an extensive dataset on North American champion trees.

Response: Again, thank you for highlighting this important work. We agree that it's really important to link our finding back to earlier work that set the foundations for our current understanding of the biophysical constraints that shape tree architecture. The papers by King 1981 and Givnish 1995 are particularly relevant to the opening two paragraphs of the introduction, where we provide a broad overview of the various constraints that trees operate under when expanding their crowns both vertically and horizontally. We have also incorporated them in Table 1, where they help support our hypotheses about how we might expect tree height and crown width to vary in relation to competition for light with neighbours. Finally, they help support several of our key findings, and we have therefore referred to them specifically in the revised discussion.

My response: Again, the authors did not – so far as I can see – insert any statement addressing the importance of these papers and the ideas and data they present for setting relative crown width. The second sentence I proposed adding (see point 1) would remedy that.

3. Finally, I fundamentally disagree with the authors' explanation for why tree height should increase with leaf nitrogen content (lines 505-508). They ignore the effect of increased photosynthesis per unit leaf mass (which increases with leaf N content) on maximum tree height. They ignore the effect of lowered allocation to roots in causing greater tree height on moister or more fertile sites AND on increasing whole-plant shade tolerance (see reference 7). The latter undercuts their argument for increased allocation to height growth ... and, of course, fails to account for decreased whole-plant shade tolerance with height (a key point made by reference 7). The authors have no data showing an increased *investment* in height growth at a given height on sites with higher leaf N content. Citing our paper to implicitly endorse their argument makes me uncomfortable.

To cope with these problems, I recommend simply re-writing the problematic lines 505-508 as follows:

Based on this we would expect species with higher leaf nitrogen to invest more in both height growth and crown expansion to allow them to intercept more light, limit self-shading and optimise the distribution of leaves across their crowns. In addition, higher leaf nitrogen

content should lead to greater photosynthetic rates and result in greater maximum tree height – as found previously along a gradient of increasing rainfall in Australia[^].

A again is Givnish et al. 2015 (see above).

If the authors make the two changes listed above, under points 1 and 3, I would fully support publication of their important manuscript.